# Summer atmospheric circulation over Greenland in response to Arctic amplification and diminished spring snow cover

Jonathon R. Preece [1] ✉, Thomas L. Mote [1], Judah Cohen [2,3], Lori J. Wachowicz[1], John A. Knox[1], Marco Tedesco[4,5,6] & Gabriel J. Kooperman[1]

The exceptional atmospheric conditions that have accelerated Greenland Ice Sheet mass loss in recent decades have been repeatedly recognized as a possible dynamical response to Arctic amplification. Here, we present evidence of two potentially synergistic mechanisms linking high-latitude warming to the observed increase in Greenland blocking. Consistent with a prominent hypothesis associating Arctic amplification and persistent weather extremes, we show that the summer atmospheric circulation over the North Atlantic has become wavier and link this wavier flow to more prevalent Greenland blocking. While a concomitant decline in terrestrial snow cover has likely contributed to this mechanism by further amplifying warming at high latitudes, we also show that there is a direct stationary Rossby wave response to low spring North American snow cover that enforces an anomalous anticyclone over Greenland, thus helping to anchor the ridge over Greenland in this wavier atmospheric state.

Summer atmospheric circulation in the Northern Hemisphere has undergone a pronounced shift in recent decades. Nowhere has this been more apparent than over the North Atlantic. Observations have revealed a more prevalent negative phase of the North Atlantic Oscillation (NAO) since the turn of the century and a coincident increase in the frequency of anomalous quasi-stationary anticyclonic conditions—known as atmospheric blocking—over Greenland[1–3].

Greenland blocking episodes transport warm, moist air from lower latitudes, which promotes widespread melt of the Greenland Ice Sheet (GrIS) through a spatially variable surface energy balance response and consequent positive feedback mechanisms[4–7]. Indeed, this dynamical forcing of the GrIS has been a primary cause of accelerating surface runoff in recent decades[1,2,8,9] and it has been estimated that a continuation of these more frequent anticyclonic conditions would result in approximately twice the surface mass loss projected by

global climate models[10], which collectively fail to capture the observed shift in atmospheric circulation over Greenland[2,11,12]. This bias has clear far-reaching implications, as it has caused even the upper range of climate projections to underestimate the contribution of GrIS surface runoff to global sea-level rise[13]. Therefore, it is of critical importance to understand whether these observed changes are merely a temporary manifestation of internal variability or a continuing consequence of anthropogenic climate change.

One active area of research aims to determine whether Arctic Amplification (AA)—i.e., an elevated rate of warming at high latitudes relative to low latitudes under global climate change—causes more frequent persistent weather extremes, including both winter cold spells[14] and summer heatwaves[15]. A prominent hypothesis underlying this area of research contends that the reduced meridional temperature gradient caused by AA results in weaker westerlies, as dictated by

[1]Department of Geography, University of Georgia, Athens, GA, USA. [2]Atmospheric and Environmental Research Inc., Lexington, MA, USA. [3]Massachusetts Institute of Technology, Cambridge, MA, USA. [4]Lamont-Doherty Earth Observatory, Columbia University, Palisades, NY, USA. [5]NASA Goddard Institute for Space Studies, New York, NY, USA. [6]Data Science Institute, Columbia University, New York, USA. ✉e-mail: jonathon.preece@uga.edu

the thermal wind relation, and a higher amplitude—or wavier—jet that propagates downstream more slowly, thus favoring persistent extremes such as those associated with atmospheric blocking[16]. While immediate follow-up studies argued that initial results supporting this hypothesis were likely an artificial byproduct of methodology[17,18], subsequent efforts have applied more robust measures of waviness, yielding statistically significant positive trends over regional domains spanning North America in summer[19,20], and the North Atlantic in both summer and winter[21]. However, previous studies have not focused on Greenland, as the regions examined have tended to dissect Greenland itself.

In what appears to be both a symptom and source of AA, there have been dramatic losses of both sea ice and snow cover since the second half of the 20th century[14,22–24]. Cryospheric response to a warming climate constitutes a positive feedback mechanism that is one of the earliest recognized sources of AA—global warming leads to retreating snow and ice cover, which lowers the surface albedo, allowing for greater absorbed shortwave radiation and further warming at high latitudes[25,26]. Thus, diminishing sea ice and snow cover

represent an important contribution to the proposed mechanism of increased waviness under AA.

Retreating snow and ice cover under AA may also impact atmospheric circulation through more direct surface-atmosphere coupling. Regional changes in snow or ice cover can elicit a stationary Rossby wave train through anomalous diabatic heating of the lower troposphere[27]. Such a stationary wave response has been documented during winter in association with sea ice variability in the Sea of Okhotsk[28] as well as the Barents and Kara seas[29]. In summer, however, the thermal gradient between the ocean and overlying atmosphere—and thus the flux of energy between them—is minimized[30]. Furthermore, the summer atmosphere is more sensitive to changes in terrestrial snow cover and associated soil moisture anomalies[15], and it has been suggested that the rapid decline of spring snow cover under AA may be a primary contributor to the recent shift in summer atmospheric circulation[31].

Spring snow cover extent (SCE) has been shown to influence summer atmospheric circulation through its residual impact on soil moisture. In what is referred to as the snow-hydrological effect, anomalously low spring SCE leads to earlier depletion of soil moisture[32], resulting in greater sensible heating of the atmosphere in summer[33–35], which can then influence the generation of Rossby waves[36–38]. Indeed, early snowmelt over Eurasia has been linked to a stationary wave response that persists through summer with increased blocking over eastern Siberia and an amplified ridge of high pressure over southeast Greenland in September[33]. These results suggest that spring SCE may exert a delayed influence on atmospheric circulation over Greenland; however, there has not yet been a direct investigation of possible teleconnections linking snow cover variability and Greenland blocking.

Here, we first document trends in atmospheric waviness spanning a Greenland-centered North Atlantic domain. We then look for evidence of a direct stationary wave response to regional SCE anomalies as a potential explanation for more prevalent anticyclonic conditions over Greenland in summer. In examining these two phenomena together, we explore the hypothesis that AA has caused an increase in waviness over the North Atlantic and that, under this wavier flow regime, the stationary wave response to concomitant reductions in SCE has supported high-pressure ridging, specifically, over Greenland.

## Results

### Summer sinuosity over the North Atlantic and its relation to Greenland blocking

To quantify the waviness of the upper-level flow in the region, we employ the sinuosity index[19,21] over a Greenland-centered North Atlantic domain spanning 0–90°W (Fig. 1). The sinuosity index is analogous to the index of the same name used in fluvial geomorphology that measures the degree to which a stream meanders by taking the ratio of the length along the stream bed to the valley, or straight-line, length[19]. When applied to the atmosphere, sinuosity measures the meanders of a given isohypse ($S_i$) and atmospheric waviness can be quantified using the aggregate sinuosity ($S_{ag}$) of a selection of isohypses spanning the midlatitudes[19].

Over the 1979–2022 study period, JJA is the only standard meteorological season with a statistically discernible trend in $S_{ag}$ (Supplementary Table 1). A linear regression of standardized JJA $S_{ag}$ against time shows that the mean $S_{ag}$ in the region has increased by 0.29 standard deviations (SD) per decade ($p = 0.014$, 95% CI [0.07 SD, 0.51 SD]). In addition to the seasonal mean $S_{ag}$, we also calculated the 25th, 50th, 75th, and 90th percentiles of daily $S_{ag}$ for each season and year. A linear regression of each percentile likewise reveals that JJA is the only season that displays a significant trend in waviness across any part of the distribution (Supplementary Table 1). Figure 2a displays the interannual variability and long-term behavior of JJA $S_{ag}$ alongside that of the Greenland Blocking Index (GBI)[39,40]. Sinuosity over the North

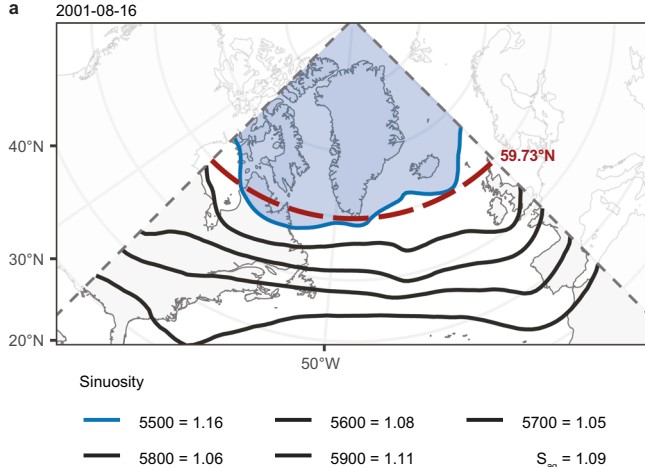

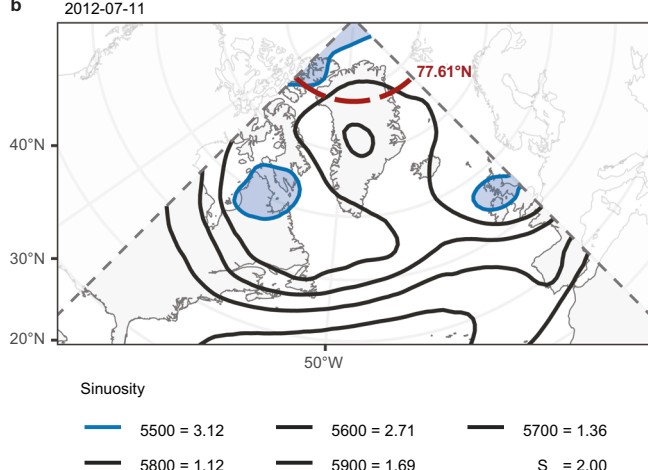

**Fig. 1 | Measuring the waviness of atmospheric circulation over the North Atlantic.** Maps illustrate the synoptic setting and aggregate sinuosity ($S_{ag}$) parameters for **a** a case of low $S_{ag}$ occurring August 16, 2001, and **b** a case of high Sag occurring July 11, 2012. The value of $S_{ag}$ as well as the sinuosity of each individual isohypse is listed at the bottom of each map. For each case, the calculation for the isohypse of maximum sinuosity (blue line) is shown as an example, where the blue shading shows the area used to calculate the equivalent latitude (red dashed line). Dashed gray lines show the longitude boundaries of the regional domain.

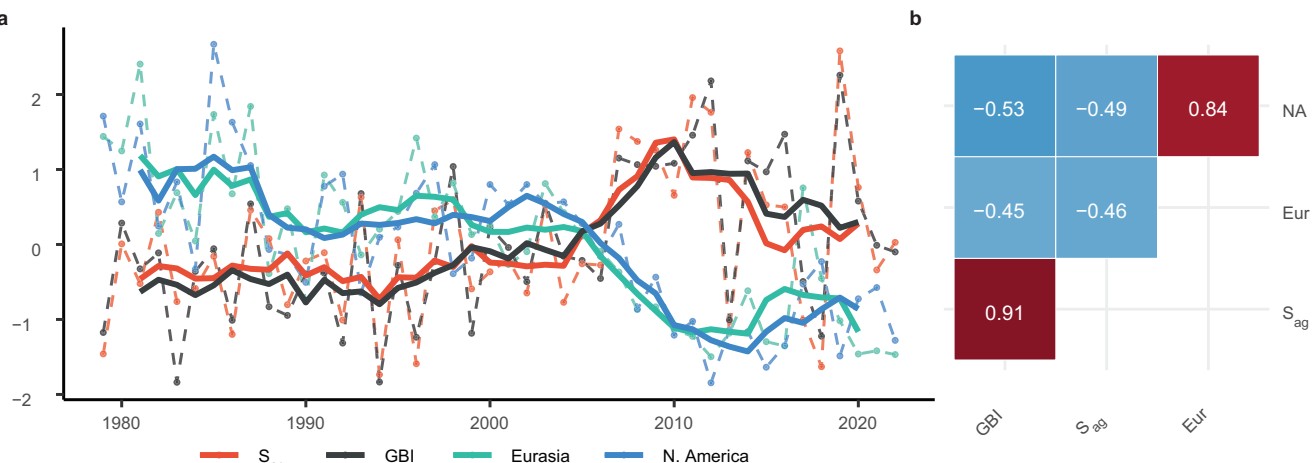

**Fig. 2 | Temporal comparison of June snow cover extent (SCE) and summer circulation over Greenland. a** Time series of June–August aggregate sinuosity ($S_{ag}$) (orange) and Greenland Blocking Index (GBI; black) plotted alongside June North American (blue) and Eurasian (green) SCE area. Dashed lines show raw annual data. Solid lines show 5-year running means. **b** Correlation matrix comparing each of the raw annual time series in (**a**) (i.e., before taking a running mean and without detrending). White numbering presents the correlation coefficient for each variable pair and color shading corresponds to the strength of correlation. NA denotes North American SCE; Eur denotes Eurasian SCE. All correlations are statistically significant at the 95% confidence level.

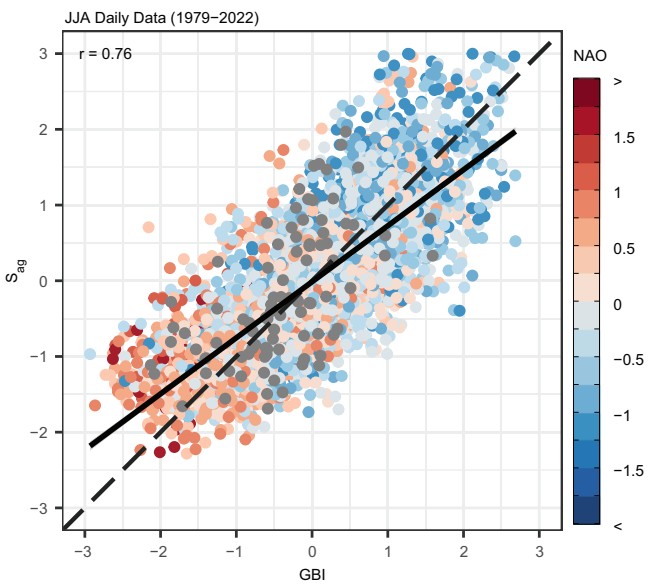

**Fig. 3 | The relationship between atmospheric waviness and anticyclonic circulation over Greenland.** Daily standardized values of aggregate sinuosity ($S_{ag}$) for days occurring during June–August (JJA) plotted against the corresponding standardized values of the Greenland Blocking Index (GBI). Dashed line marks the 1:1 line. Solid line shows the linear best-fit line. Data points are color-coded according to the corresponding monthly value of the principle-component-based North Atlantic Oscillation (NAO) index[45] as shown in the color bar to the right. Gray points correspond to data from 2022, for which NAO index values are not yet available. The associated Pearson's correlation coefficient ($r$) is provided in the upper-left corner.

Atlantic and Greenland blocking exhibit similar long-term behavior, with a strong increase from the mid-1990s through the early 2010s followed by high variability thereafter. Indeed, the linear trend estimated from the annual JJA GBI time series indicates a statistically significant increase of 0.37 SD per decade ($p = 0.001$, 95% CI [0.15 SD, 0.58 SD]), comparable to $S_{ag}$. The concurrence between $S_{ag}$ and the GBI is also apparent over interannual timescales (Fig. 2a). This, combined with the shared long-term positive trend, results in a strong positive correlation between the two annual time series of $r = 0.91$

(Fig. 2b). Focusing on each summer month separately, the greatest increase in both the $S_{ag}$ and the GBI has occurred during the months of July and August (Supplementary Fig. 1).

That there is a strong correlation between $S_{ag}$ and the GBI over seasonal timescales is not surprising given the well-established increase in summer Greenland blocking in recent decades and that blocked conditions represent an anomalously meridional—i.e., wavy—circulation pattern. Thus, the increase in blocking frequency and/or magnitude signaled by an above-normal GBI, either over the course of one summer or over the long term, would by definition act to increase North Atlantic sinuosity over the same period. However, the reverse may not necessarily be true—i.e., an increase in North Atlantic sinuosity could translate to an anomalously low GBI if the wavy circulation pattern places a trough over Greenland rather than a ridge. To explore this relationship more broadly, we examine the correlation between the two variables over daily timescales by plotting standardized values of $S_{ag}$ against the corresponding standardized values of the GBI in Fig. 3. We also display the associated monthly NAO index value for each data point. As expected from previous work on the relationship between the NAO and Greenland blocking[3], anomalously high $S_{ag}$ and GBI tend to occur under the negative phase of the NAO, while the opposite conditions prevail under its positive phase. Critically, there is a clear linear relationship between $S_{ag}$ and the GBI, and they are highly positively correlated at $r = 0.77$, signaling a strong tendency for wavy circulation over the North Atlantic to coincide with anticyclonic conditions over Greenland. These results are consistent with the view that interannual variability in the NAO is simply a reflection of two principal atmospheric states over the North Atlantic—a blocked state with an anomalous anticyclone over Greenland that weakens the westerly flow and manifests as the negative phase of the NAO, and an unblocked state with strong westerly flow that manifests as the positive phase of the NAO[41].

The above results show an interrelated increase in waviness over the North Atlantic and blocking over Greenland in recent decades. To investigate how these changes may be a feature of Arctic amplification, we followed previous work[19] and calculated sinuosity as a function of latitude, then calculated seasonal trends in sinuosity at each latitude using a rolling linear regression. Figure 4 depicts the moving linear trend in the 500 hPa sinuosity over the North Atlantic alongside that for the 850 hPa meridional temperature gradient and the 500 hPa

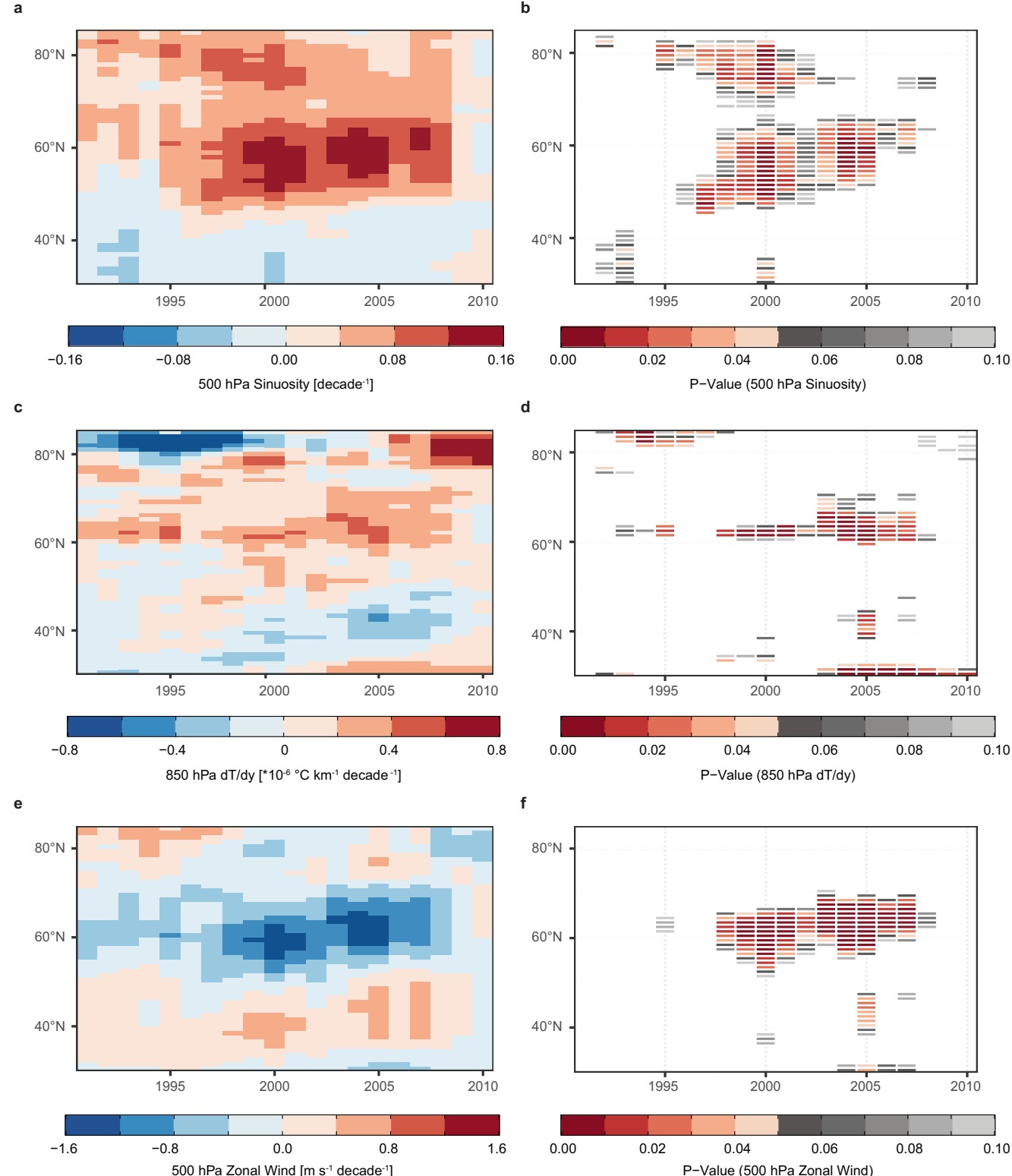

**Fig. 4 | Linear trends in atmospheric circulation over the North Atlantic.**
Shading in **a**, **c**, **e** denotes the slope coefficient of a rolling linear regression of **a** 500 hPa sinuosity, **c** 850 hPa meridional temperature gradient, and **e** 500 hPa zonal wind against time over a 25-year period centered on the year specified along the x-axis and at each latitude listed along the y-axis. Shading in **b**, **d**, **f** shows the corresponding *p*-values for the slope estimates of **b** 500 hPa sinuosity, **d** 850 hPa meridional temperature gradient, and **f** 500 hPa zonal wind. Hypothesis tests were conducted individually for each time window, with no multiple testing adjustment. All quantities calculated for the North Atlantic domain (0–90°W).

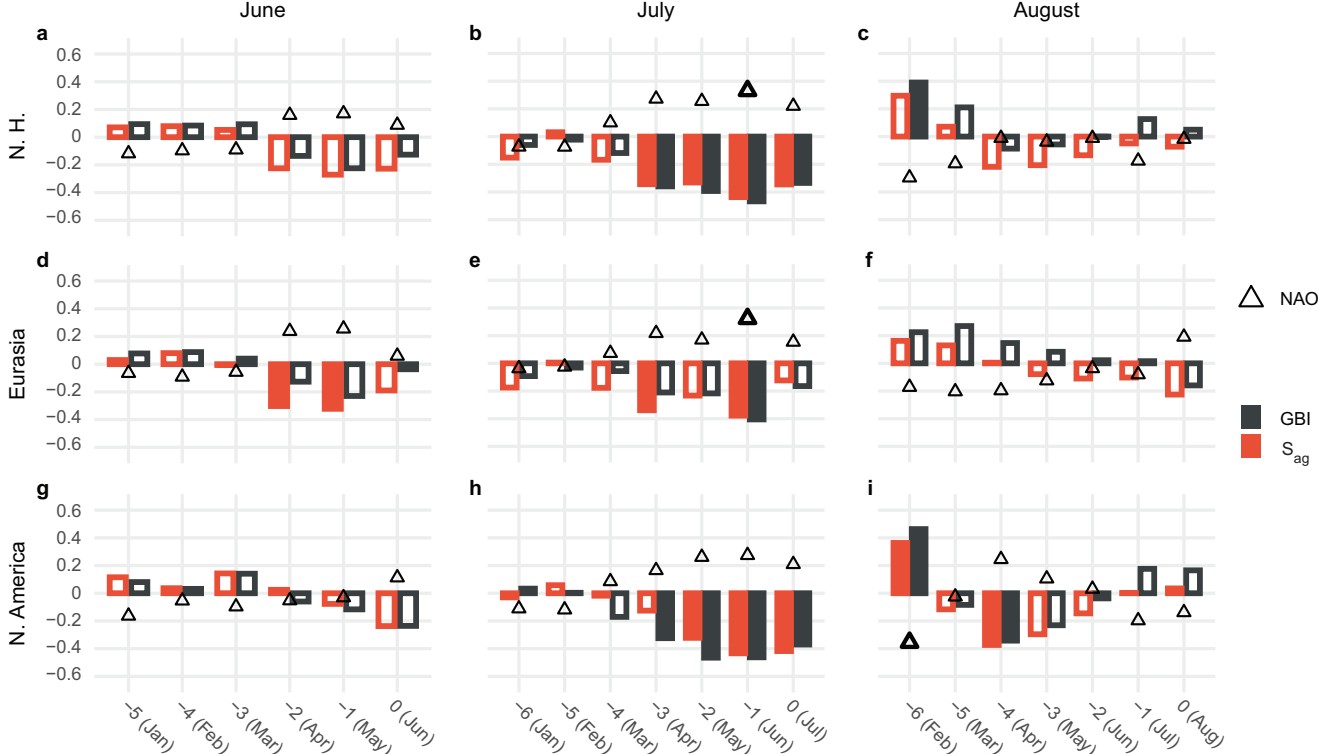

**Fig. 5 | Relationship between summer atmospheric circulation over Greenland and antecedent snow cover extent.** Panels present lagged correlation plots relating **a**, **d**, **g** June, **b**, **e**, **h** July, and **c**, **f**, **i** August atmospheric circulation over Greenland to antecedent monthly snow cover extent (SCE) area over **a**–**c** the Northern Hemisphere, **d**–**f** Eurasia, and **g**–**i** North America. The color-coded bars in each panel document the correlation coefficient relating the Greenland Blocking Index (GBI; black), and aggregate sinuosity ($S_{ag}$; orange) measured during the summer month labeled at the top of each column to lagged monthly SCE area as noted along the x-axis. Likewise, triangles note the correlation between the principal-component-based North Atlantic Oscillation (NAO) index[45] and lagged monthly SCE area. Filled bars and bold triangles indicate statistical significance at $\alpha = 0.05$. Hypothesis tests were conducted individually for each time lag, with no multiple testing adjustment.

zonal mean zonal wind. In each subplot, the change in shading in the x-direction documents the slope of the linear trend at each latitude over a 25-year time window centered on the year that is indicated along the x-axis.

Focusing on the change in sinuosity (Fig. 4a, b), it is clear that the positive trend in JJA $S_{ag}$ evident in Fig. 2a has been driven by changes at higher latitudes. The most pronounced trend has occurred around 60°N; however, strong increases in sinuosity accompanied by low **p**-values (Fig. 4b) are also evident extending poleward through 80°N. This is in close agreement with the latitudinal distribution of trends in sinuosity over a North America domain documented in previous work[19]. Temporally, the slope of the trend is greatest when the regression is performed from ~1995 onward, coinciding with the upturn in the 5-year running mean in both $S_{ag}$ and the GBI visible in Fig. 2a. The slope of the regression line approaches zero for time windows centered on years beyond 2008, consistent with the reduction in $S_{ag}$ in the latter part of the study period (Fig. 2a). However, even with the decline in waviness relative to its peak around 2010, the 5-year running mean of $S_{ag}$ remains higher than at any point prior to the turn of the century (Fig. 2a).

Figure 4b shows a decrease in the strength of the meridional temperature gradient between 60° and 70°N. As expected from the thermal wind relation, the weakened temperature gradient in the lower troposphere is met by a reduction in the zonal mean zonal wind aloft (Fig. 4c). In each case the strongest trend aligns with the location (~60°N) and timing of the observed change in sinuosity. Thus, the conditions accompanying the increase in North Atlantic sinuosity and Greenland blocking in recent decades are congruent with a leading theoretical framework[16] of midlatitude circulation change under AA.

### Northern Hemisphere snow cover extent and atmospheric circulation over Greenland

Figure 2a also displays the long-term variability in June North American and Eurasian SCE. A linear fit to the annual values of June SCE indicates a decline of −0.56 SD ($p = 0.000$, 95% CI [−0.73 SD, −0.39 SD]) and −0.59 SD ($p = 0.000$, 95% CI [−0.75 SD, −0.43 SD]) per decade over North America and Eurasia, respectively—trends that mirror that of $S_{ag}$ and the GBI. Previous work has highlighted June conditions in particular as an effective indicator of spring snow cover variability more broadly[42]. Thus, these results indicate that the seasonal timing of this pronounced change in surface conditions precedes that of the observed change in atmospheric conditions, suggesting the possibility of a causal relationship.

The negative correlations relating both North American and Eurasian spring SCE to summer $S_{ag}$ and GBI presented in Fig. 2b may largely reflect the inverse long-term trends evident in their respective 5-year running means. To investigate the matter further, Fig. 5 presents the results of a lagged regression analysis comparing each of the three circulation indices with antecedent spring SCE. Here, we linearly detrend each time series before conducting the regression to more aptly focus on lagged covariance over interannual timescales.

The top row of Fig. 5 details the lagged relationship between antecedent SCE area over the Northern Hemisphere as a whole and June, July, and August monthly atmospheric circulation over Greenland. Partitioning summer into its constituent months reveals a clear negative lagged correlation signature relating July $S_{ag}$ and GBI to spring Northern Hemisphere SCE, with significant correlations extending as far back as the preceding April. The sign of the lagged correlation indicates that low spring Northern Hemisphere SCE is typically

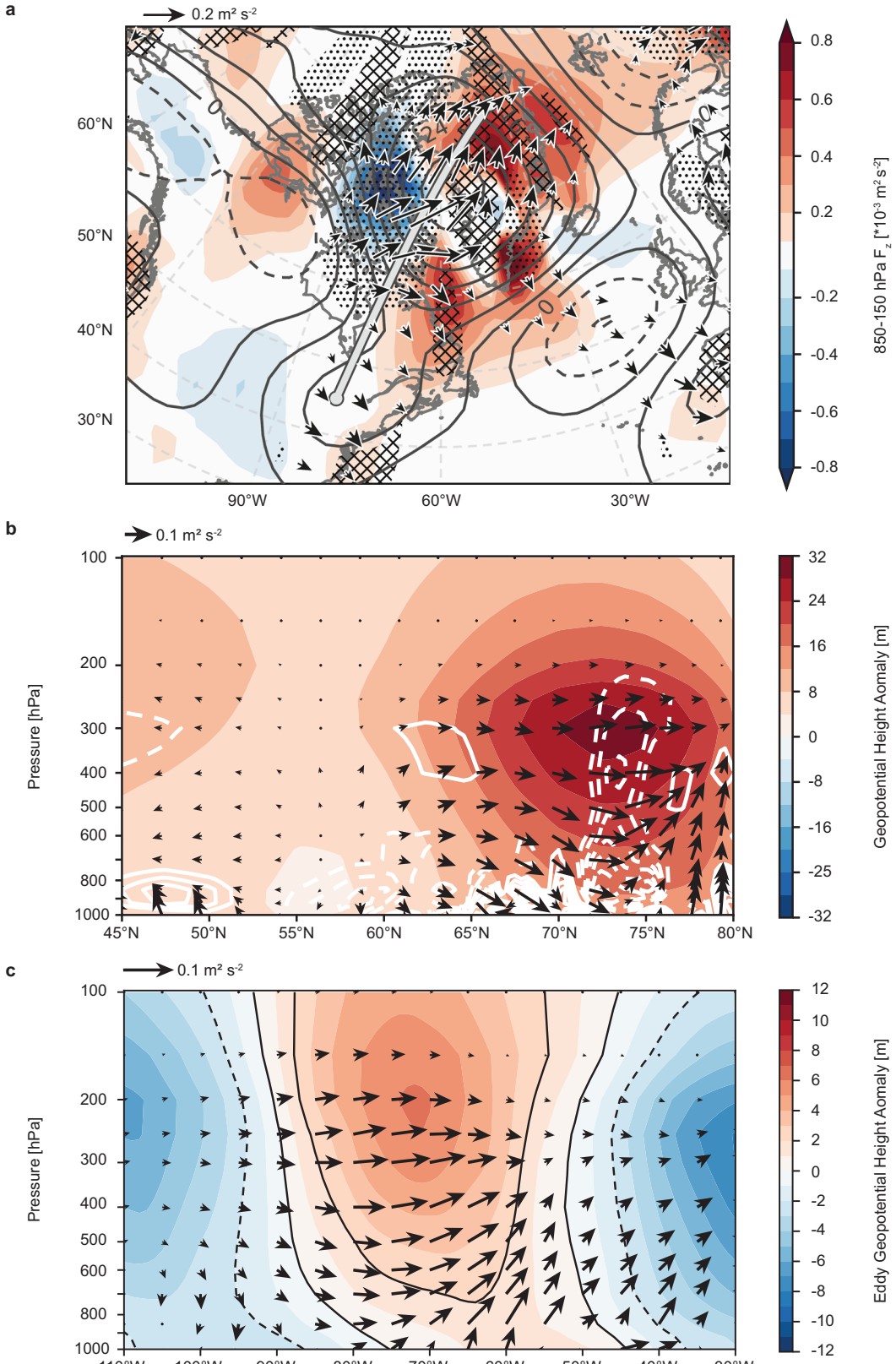

followed by an increase in July sinuosity over the North Atlantic and an attendant increase in Greenland blocking. This negative lagged correlation signature is mirrored by a positive correlation between the principal-component-based NAO index of Hurrell[43] and antecedent SCE; however, only the relationship with June SCE is significant at $\alpha = 0.05$. The lagged regression plots for June and August atmospheric conditions show no apparent lagged relationship with spring SCE (Fig. 5).

Considering SCE area over North America and Eurasia individually likewise suggests a lack of a lagged correlation between spring SCE and either June or August circulation over Greenland (Fig. 5). There is a positive correlation between February SCE over North America and

**Fig. 6 | Stationary Rossby wave response to low North American snow cover extent (SCE). a** Contours display the 300 hPa geopotential height (Z300) response to one standardized negative anomaly of 1979–2022 North American SCE as estimated from a linear regression performed at each grid cell. Contour interval: 4 hPa (solid, positive; dashed, negative). Vectors indicate the 300 hPa horizontal component and shading indicates the average 850–150 hPa vertical component of the attendant wave activity flux (WAF), which was derived from the Z300 regression coefficients shown by the contours. Stippling (crosshatching) indicates horizontal divergence (convergence) of the 300 hPa WAF exceeding $\pm 10^{-7}$ m s². **b** Vertical cross-section along the thick gray line in (**a**), where vectors indicate WAF, shading indicates the geopotential height response, and solid (dashed) white contours indicate WAF divergence (convergence). Contour interval: $10^{-7}$ m s², zero contour omitted. The horizontal component of the vectors in (**b**) is 100 times that of the vertical component. **c** Vertical cross-section of the mean 40–60°N eddy geopotential height response (shading) obtained from the coefficients of a linear regression against standardized, inverted May North American SCE area. The −2, 0, and 2 hPa isopleths are shown in black to more clearly visualize the vertical structure of the eddy height anomalies. Vectors show the meridional average of the associated WAF.

August circulation over Greenland (Fig. 5i); however, given the strong seasonality of snow cover, the isolated nature of this result does not present compelling evidence of a lagged relationship, and a plausible physical mechanism underlying such a relationship is not immediately clear. Examining the two continental regions separately does, however, clarify that the lagged signature evident for the Northern Hemisphere as a whole is primarily driven by a relationship with snow cover over North America (Fig. 5h).

It appears the significant correlation between the July NAO and June Northern Hemisphere SCE (Fig. 5b) is primarily a reflection of the relationship with SCE over Eurasia (Fig. 5e). This result is consistent with previous work showing that there is an increase in the poleward propagation of Rossby waves in years of low June Eurasian SCE that acts to enforce the negative phase of the Northern Annular Mode during summer months[42]—an atmospheric state which supports more frequent high-latitude blocking, including over Greenland[2,3,44]. Previous work has also shown that spring Eurasian SCE exerts a far-reaching influence on the summer stationary wave pattern[33]; however, this previous analysis did not inform on July conditions over Greenland. While Eurasian SCE may also play a role, the results presented in Fig. 5 clearly show a more robust relationship with spring snow cover over North America.

One way that retreating spring SCE may favor Greenland blocking is by amplifying the rate of warming at high latitudes and reducing the meridional temperature gradient[31,45], thereby contributing to the mechanisms examined in the previous section. Another possibility that we explore here is that low SCE may act to enforce ridging over Greenland via a direct stationary Rossby wave response. Figure 6 traces the July stationary wave response to low May North American SCE using the three-dimensional wave activity flux (WAF)[46]. We focus on the relationship with May SCE because that is the month that exhibits the earliest significant lagged relationship with July observations of both GBI and $S_{ag}$ (Fig. 5h).

Figure 6a presents the linear regression coefficients relating July 300 hPa geopotential height (Z300) to standardized May North American SCE area. Here, we inverted the SCE time series and detrended all data before conducting the regression to emphasize the response to low SCE. The Z300 isopleths indicate that years of low SCE are followed by upper-level ridging over much of northeastern North America and Greenland, with a strong anomalous high-pressure center located over Baffin Bay and a weaker high over the northern Appalachians. Repeating this regression for several 20-year periods within our 1979–2022 study period shows that this upper-level pattern is consistent throughout the study period and is robust to changes to the start and end years (Supplementary Fig. 2). Figure 6a also depicts the vertical (shading) and horizontal (vectors) components of the associated WAF and its horizontal divergence (stippling). The origins of the upper-level response appear to be connected to an area of upward WAF over Eastern Canada, with the horizontal WAF indicating that the ensuant stationary wave activity propagates northeast from the area around Hudson Bay before converging over Greenland. Convergence of stationary wave activity is associated with the poleward advection of low potential vorticity and has been shown to be a key contributor to the development of blocked flows[46,47]. Thus, WAF convergence over

western Greenland is supportive of increased blocking in years of low spring SCE.

To further investigate the ridging over Greenland, Fig. 6b plots a vertical cross-section of the anomalous geopotential height and associated WAF along the solid gray line in Fig. 6a. The vertical cross-section more precisely shows upward WAF in the lower troposphere between 54° and 62°N. The WAF vectors then turn poleward but are trapped by the tropopause, with convergence enforcing the strong high-pressure anomaly over Baffin Bay and western Greenland that extends throughout the depth of the troposphere.

The conditions that give rise to this Rossby wave activity are apparent in Fig. 6c, which presents the vertical structure of the July eddy geopotential height anomaly and associated WAF averaged over 40–60°N. Like the geopotential height anomalies in Fig. 6, the anomaly field here was estimated from a linear regression of July eddy geopotential height against standardized, inverted May North American SCE area. The westward tilt of the eddy geopotential heights with altitude is characteristic of a baroclinic structure of the atmospheric column and is strongest in the lower levels of the troposphere. This westward tilt in the eddy geopotential height field supports the upward propagation of Rossby waves[48,49] that then converge over Greenland, thereby enforcing the anomalous ridge. In the section "The snow-hydrological effect as a bridge between spring snow cover variability and summer atmospheric response", we consider the snow-hydrological effect as an explanatory mechanism behind this delayed baroclinic response.

## Preliminary GCM results

Preliminary results from a 10-year global climate model (GCM) simulation in which we removed all snow cover over the whole of North America on May 1st of each year closely reproduces the anticyclonic anomaly over Baffin Bay and the attendant WAF from over Hudson Bay and Eastern Canada; however, as opposed to the July response that is evident in observations the stationary wave response in the model emerges a month earlier in June (Supplementary Fig. 3b). The early appearance of the stationary wave response relative to observations may be due to the abrupt reduction in snow cover that we imposed in the model—a possibility that we will explore in future work. Regardless, the strong agreement between the modeled and observed atmospheric response presents robust support of a physical link between low spring North American snow cover and more prevalent anticyclonic circulation over Greenland in summer.

## The snow-hydrological effect as a bridge between spring snow cover variability and summer atmospheric response

The snow-hydrological effect represents a causal chain of delayed surface-atmosphere coupling in which low spring snow cover leads to earlier depletion of soil moisture and a consequent increase in surface sensible heating that locally warms the lower troposphere in summer[33,34]. To investigate the role that this snow-hydrological effect may play in the observed July stationary wave response, we first delineate the climatological retreat of spring North American SCE and spatially resolve the relationship between spring snow cover and July Greenland blocking in Fig. 7. Next, we illustrate the spatiotemporal

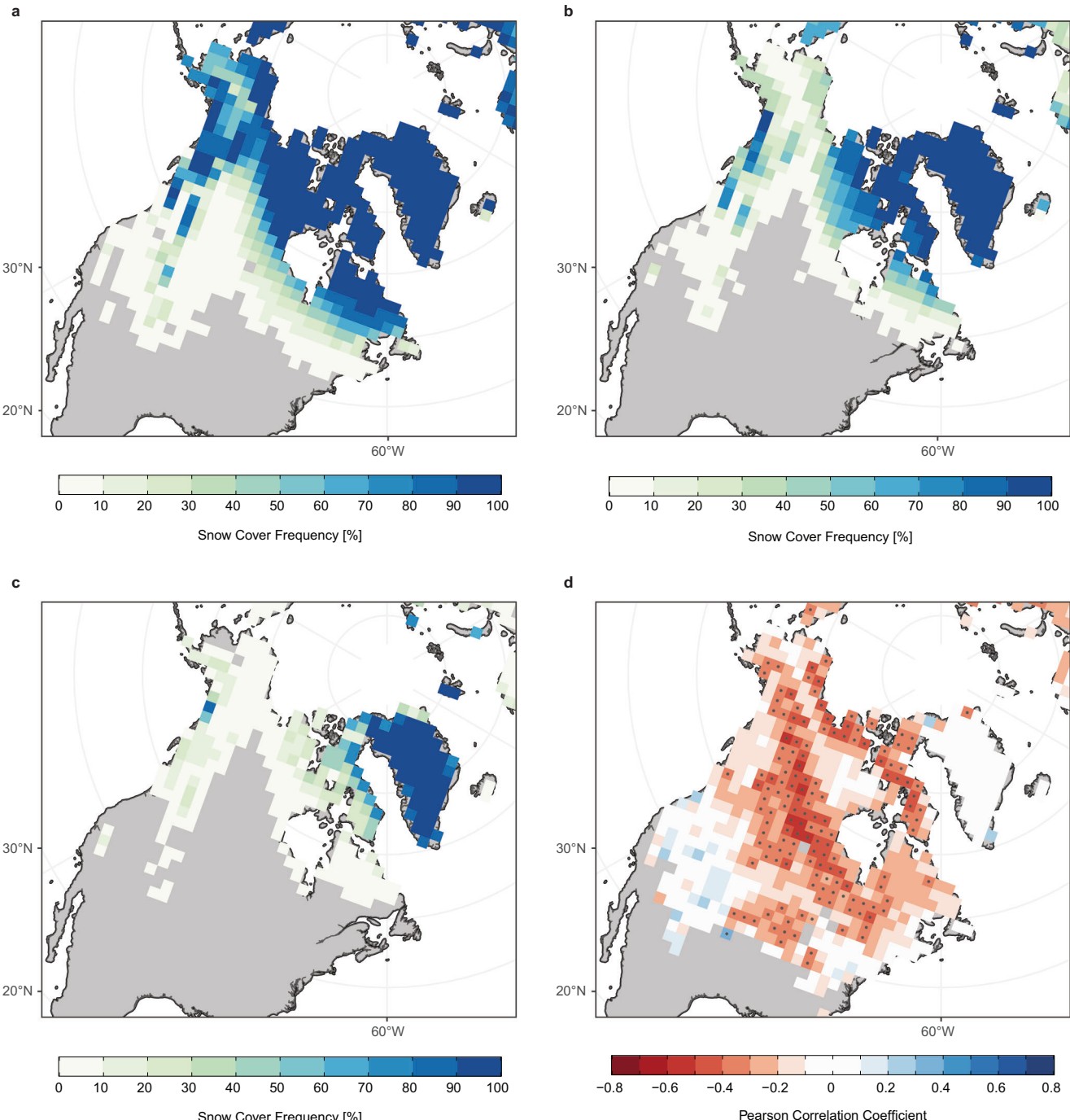

**Fig. 7 | North American snow cover extent (SCE) variability and its relationship with Greenland blocking.** Panels **a**–**c** show the 1979–2022 frequency of weekly snow cover at each grid point during **a** May, **b** June, and **c** July. Panel **d** presents grid point Pearson correlation coefficients relating the total number of weeks with snow cover from April to June of each year and the monthly mean Greenland Blocking Index for the following July. Note that negative values are shaded red to reflect an increase in blocking associated with a decrease in snow cover duration. Snow cover is represented using the National Oceanic and Atmospheric Administration SCE Climate Data Record[64]. Stippling indicates statistical significance at the 95% confidence level. Hypothesis tests conducted individually for each grid cell without multiple testing or spatial autocorrelation correction.

progression of North American soil moisture and surface temperature anomalies in response to low May SCE in Fig. 8 and discuss these results in the context of the observed July stationary wave response.

Figure 7a–c presents the frequency of observed snow cover from 1979 to 2022 for each month spanning the lagged relationship between May SCE area and July circulation over Greenland. In May, there is a region of unremitting snow cover extending from the Arctic coast to the southern end of Hudson Bay (Fig. 7a). South of this point,

there is a zonal band of snow cover frequencies in the range of 30–70% marking the primary area of May SCE variability. Snow cover frequencies greater than 50% in June are generally confined to the Canadian Arctic tundra and the mountainous areas along the Pacific coast (Fig. 7b). By July, the continent is predominately snow free (Fig. 7c).

To spatially delineate the regions of SCE variability that are linked to atmospheric circulation over Greenland, we show the grid point

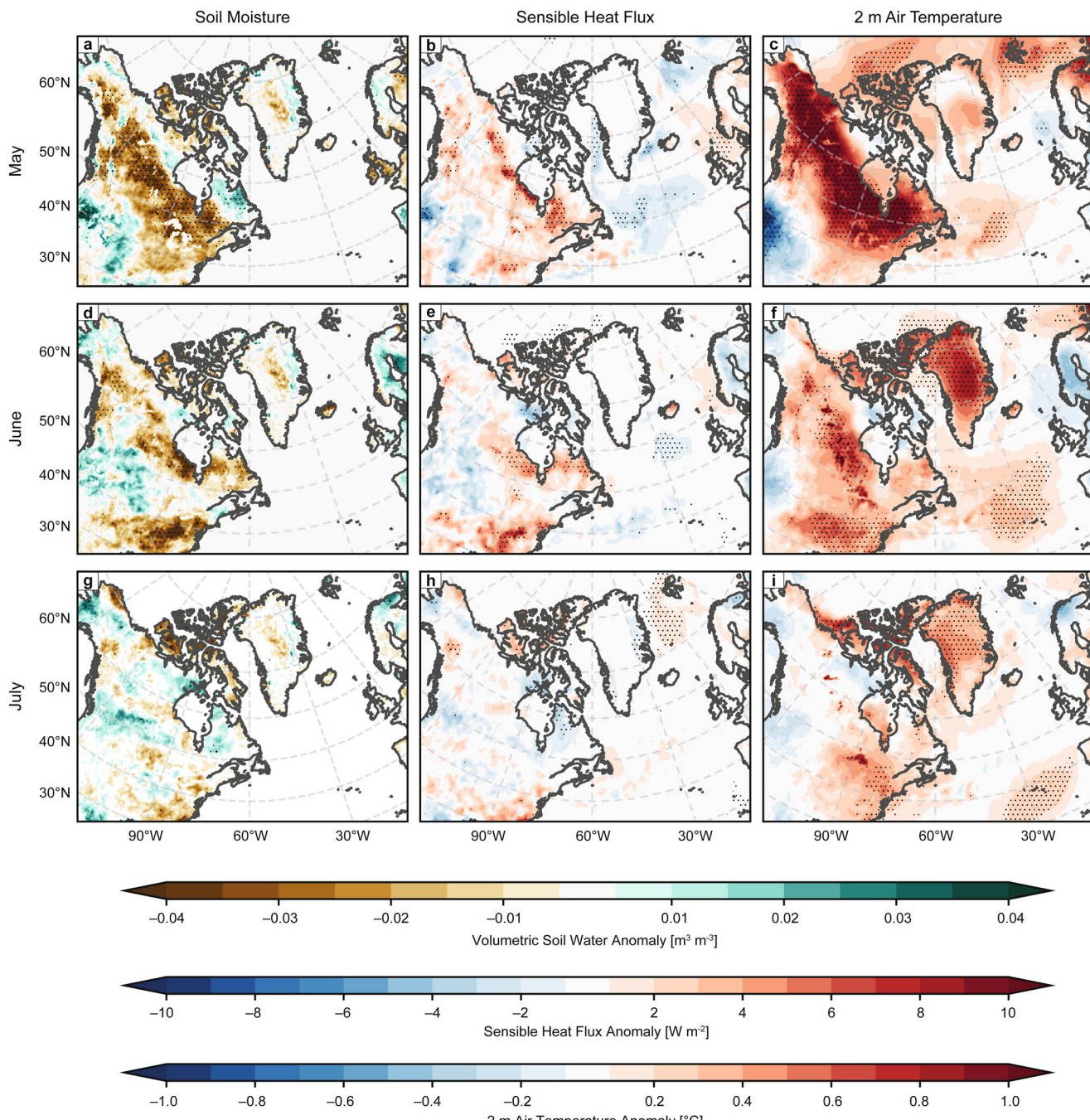

**Fig. 8 | North American surface conditions following low May snow cover extent.** Regression coefficients of **a**, **d**, **g** ERA5 volumetric soil water to a depth of 28 cm, **b**, **e**, **h** upward surface sensible heat flux, and **c**, **f**, **i** 2 m air temperature obtained from a linear regression of **a–c** May, **d–f** June, and **g–i** July surface variables against inverted and standardized May North American snow cover extent area. Stippling indicates significance at the 95% confidence level. Hypothesis tests conducted individually for each grid cell without multiple testing or spatial autocorrelation correction.

correlation between detrended July GBI and detrended spring North American snow cover duration in Fig. 7d. Here, we define spring snow cover duration as the total number of observations from April to June of each year in which the NOAA SCE CDR indicates the presence of snow in each grid cell. The NOAA SCE CDR does not inform on the fraction of snow cover within each grid cell. Since spring snow cover extent closely reflects the timing of annual snow melt[42], we used snow cover duration to construct a non-Boolean time series at each grid cell that could be compared against variability in the GBI. The strong negative correlation values stretching zonally from southern Alaska to the Labrador Peninsula and equatorward into the interior plains of the

US indicate that low spring SCE in this region is typically associated with more prominent anticyclonic conditions over Greenland in July. This region of high correlation generally aligns with the zonal band of moderate May SCE frequency in Fig. 7a and suggests that snow cover variability in this area drives the lagged correlation documented in Fig. 5.

Figure 8 chronicles the surface response to spring SCE variability by plotting anomalies of volumetric soil water content (left column), surface sensible heat flux to the atmosphere (middle column), and 2 m air temperature (right column) obtained from a linear regression against standardized, inverted May North American SCE area for each

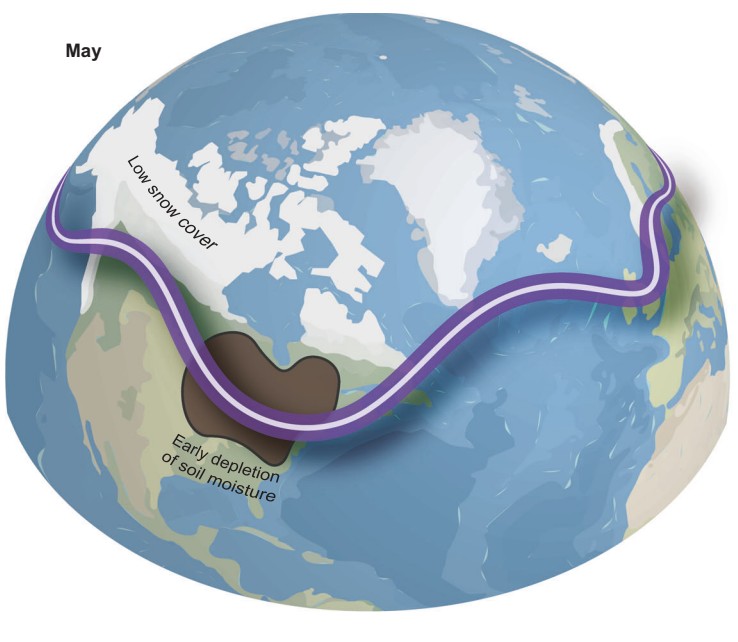

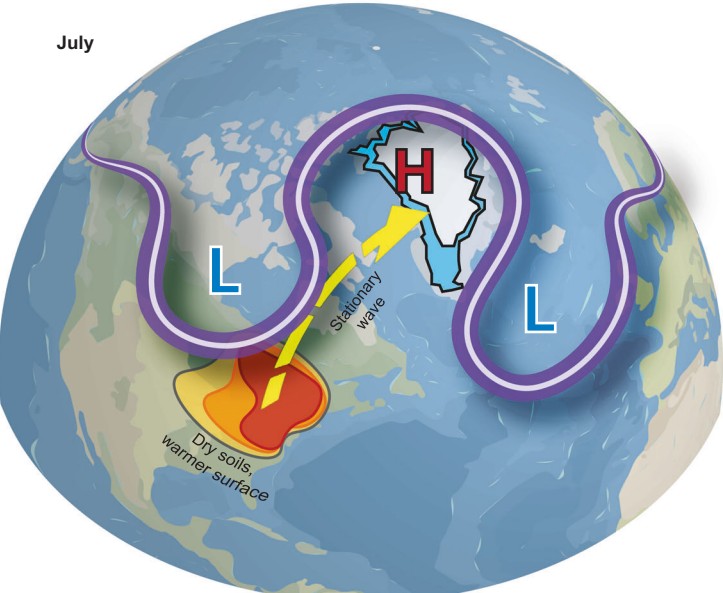

**Fig. 9 | Illustration of the link between North American snow cover and atmospheric circulation over Greenland.** Low spring snow cover causes early depletion of soil moisture over eastern North America (brown shading). The dry conditions that follow produce a warm surface anomaly that persists into July (orange shading). This anomalous heating of the near-surface atmosphere induces

a stationary Rossby wave response that propagates northeastward (yellow arrow) and favors high-pressure ridging over Greenland—conditions which are known to augment melt of the Greenland Ice Sheet (blue shading). Red H denotes high pressure; blue L denotes low pressure.

month spanning the lagged relationship identified in Fig. 5 (i.e., May–July). Again, all fields were detrended prior to the regression. The regression anomalies indicate that low May SCE area coincides with anomalously warm surface air temperature along the northern coastline of North America that extends just south of the Great Lakes (Fig. 8c). In June, the positive temperature anomalies throughout Canada have weakened considerably (Fig. 8f). At the same time, strong negative soil moisture anomalies have emerged in conjunction with an increase in sensible heat flux and a secondary warm anomaly over eastern North America (Fig. 8d–f). The emergence of this warm anomaly well after the seasonal retreat of snow cover from the region (Fig. 7) is consistent with an active snow-hydrological effect. By July, soil moisture and sensible heat flux anomalies over the whole of North America have weakened further (Fig. 8g, h); however, while not

statistically significant, the spatial pattern of reduced soil moisture in the eastern US and Canada aligns well with positive surface temperature anomalies running along the east coast and extending inland to the south of Hudson Bay that are significant at the 95% confidence level (Fig. 8g, i)—conditions which suggest that the snow-hydrological effect over eastern North America persists into July when the stationary wave response is observed.

Considering these surface conditions in the context of the concomitant stationary wave response, it is apparent that the upward WAF in Fig. 6c generally aligns with the warm anomaly center over eastern North America (Fig. 8f, i). This is consistent with previous work, which has shown that changes in both snow cover[33] and sea ice[29] can cause a local baroclinic response that favors upward WAF over the region of consequent diabatic heating of the near-surface atmosphere. Thus,

these results support the existence of a regionally focused snow-hydrological effect over the eastern North America that persists into July, providing a chain of physical mechanisms bridging spring SCE to the baroclinically generated stationary Rossby wave response.

## Discussion

AA has often been invoked as a possible explanation for the atmospheric conditions that have accelerated GrIS mass loss in recent decades[16,31,50,51]. Our analysis directly investigated this possibility by examining the connection between AA and the waviness of atmospheric circulation over the North Atlantic, and by documenting the role of dwindling terrestrial snow cover in dictating the position of the resulting wave pattern. In pursuing this objective, we documented evidence linking AA to the observed increase in summer Greenland blocking.

Summer atmospheric sinuosity across our Greenland-centered domain has undergone a statistically significant increase that is consistent with long-term positive trends in waviness to the east and west of Greenland documented in previous studies[19–21] (Figs. 2a, 4a). The increase in sinuosity is collocated with a simultaneous weakening of the meridional temperature gradient and deceleration of the zonal wind (Fig. 4)—results that suggest AA has contributed to the increase in persistent weather extremes in the region[16].

An increase in sinuosity does not inform on any tendencies in the phase of the measured waves over a given location. A cursory look at Fig. 2 suggests that the increase in sinuosity over the North Atlantic has indeed supported the well-established positive trend in Greenland blocking. Since a blocking anticyclone is, itself, a feature of an amplified planetary wave, it is not surprising that the long-term behavior of the GBI and that of North Atlantic sinuosity have closely tracked each other (Fig. 2). However, our analysis goes further in demonstrating that sinuosity and the GBI exhibit a strong linear relationship over daily timescales (Fig. 3), thereby establishing that wavier flows in the region do, in fact, favor ridging over Greenland.

Others have summarized the dynamics that encourage ridging over Greenland more generally[3]. They note that westerly flow over the north-south oriented topography produces ridging, while katabatic outflow from the ice sheet's interior generates subsidence and adiabatic warming aloft, which increases geopotential heights. Furthermore, there is frequent cyclogenesis along the US Atlantic coast, which promotes the development of a downstream ridge over Greenland[3,52]. These factors may, on their own, anchor the ridge over Greenland during periods of wavier flow.

Given the strong linear relationship between sinuosity and the GBI, one could expect a positive trend in Greenland blocking simply by virtue of increasing waviness under AA. Retreating SCE would likely contribute to this mechanism by furthering high-latitude warming and weakening the meridional temperature gradient[31,45], but SCE anomalies are known to exert a direct influence on the stationary wave pattern[33,53,54], which could also act to enforce ridging over Greenland. Our results show a significant lagged relationship indicating that years of low spring North American SCE are followed by increased sinuosity and Greenland blocking the following July (Fig. 5). This delayed atmospheric response to spring SCE appears to be the consequence of an active snow-hydrological effect that emerges over eastern North America in June and persists into July in years of low May SCE (Fig. 8). As summarized in Fig. 9, the anomalously warm surface temperatures that follow are associated with a baroclinic atmospheric response in July that excites a stationary Rossby wave train and enforces anomalous ridging over Greenland (Fig. 6).

Our results indicate that a weakened westerly flow under AA and a stationary wave response to retreating North American SCE have both contributed to more frequent summer Greenland blocking in recent decades. These represent two potentially synergistic phenomena, as the atmospheric response to surface thermal forcing in the midlatitudes becomes more pronounced with a reduction in the background zonal flow[15,55,56]. Moreover, considering these two mechanisms together may provide additional insight into the dynamics underlying the recent increase in Greenland blocking if viewed within the traffic jam model of blocked flows[57]. Under this model, blocking is primarily governed by the column-averaged zonal flux of the local wave activity (LWA), which is a longitudinally varying, potential-vorticity-based measure of jet stream waviness[58]. Like the flow of traffic on a highway, the zonal flux increases to accommodate intensifying LWA up to some capacity, after which the flux declines rapidly, leading to the buildup of LWA and the onset of blocking[57]. According to the model, a more amplified stationary wave in response to retreating spring snow cover (Fig. 6) would act to bring the LWA closer to capacity, while a weaker jet under AA (Fig. 4c) reduces that capacity[57,59]—changes that would both lessen the transient eddy forcing required to instigate blocking over Greenland.

While the method we used to trace the stationary wave response in this analysis emphasizes the relationship between spring SCE and the atmospheric circulation that follows, the pattern and magnitude of the regression anomalies also reflect the combined influence of other sources of internal variability within the climate system. Other aspects of AA may also play a role—previous work has presented evidence of a link between declining sea ice and anomalous anticyclonic circulation over Greenland[50,60]. It is difficult to disentangle these integrated processes using observations alone. Preliminary GCM results show an upper tropospheric response to the disappearance of spring North American snow cover that closely aligns with the observational results presented herein (c.f., Fig. 6a and Supplementary Fig. 3b), increasing confidence in a link between declining snow cover and the recent shift in summer atmospheric circulation over Greenland. Future work will continue to examine the representation of this snow-atmosphere coupling in GCMs using both idealized snow-cover experiments and ensemble simulations of historical and projected conditions.

## Methods

### Characterizing the large-scale flow over Greenland

This analysis utilizes ERA5 reanalysis data[61] over a study period of 1979–2022. Here we focus on 1979 onward due to the relative scarcity of assimilated observations and issues with the initialization of soil moisture in the ERA5 dataset prior to this point[62]. We represent the large-scale atmospheric setting over Greenland as quantified using two circulation metrics: the sinuosity index and the GBI. The sinuosity of a given isohypse is calculated as follows[19]:

$$S_i(t) = \frac{L_i}{L_\phi} \tag{1}$$

Where $S_i$ is the sinuosity index for a given isohypse $i$ on day $t$, $L_i$ is the arc length along the isohypse, and $L_\phi$ is the small-circle length along the equivalent latitude. $L_\phi$ is determined as the parallel of latitude that contains an equal area on its poleward side as is enclosed by the isohypse in question. Thus, $S_i(t) = 1$ would indicate an isohypse that traces a parallel of latitude. These index parameters are illustrated in Fig. 1 where, using the 5500 hPa isohypse in Fig. 1a as an example, the blue shading shows the area that was used to calculate an equivalent latitude of 59.73°N. As noted to the right of the map, the ratio of length along the blue line to the length of the red dashed line yields a relatively low sinuosity for the 5500 hPa isohypse of 1.16.

To characterize atmospheric circulation over the North Atlantic as a whole, we apply an aggregate sinuosity, $S_{ag}(t)$, similar to that of ref. 19, that is simply the mean $S_i(t)$ of five evenly distributed isohypses covering the midlatitudes. The aggregate sinuosity for a given period

of interest (e.g., monthly or seasonal values) is then

$$S_{ag} = \frac{\sum_{t}^{T} S_{ag}(t)}{T} \qquad (2)$$

Where $T$ is the number of days during the period. Figure 2 shows the JJA $S_{ag}$ for the 1979–2022 study period.

While ref. 19 employed the same five isohypses throughout the year, we found that a seasonally defined $S_{ag}$ better captured the baroclinic latitudes, particularly in summer (see Supplementary Fig. 5). Here, we use seasonally varying $S_{ag}$ isohypses that are evenly spaced between the 1979–2022 average zonal mean Z500 at 30°N and 70°N for each standard meteorological season. From its definition, $S_i$ should not measure less than one. While this holds true for isohypses that traverse the entire domain, in certain situations where they do not— e.g., a region of high pressure covering only the southeast corner of the domain—$S_i$ can assume an unrealistically low value. Thus, we take the additional step of filtering all values of $S_i < 1$ before calculating $S_{ag}$.

In its atmospheric application, a low $S_{ag}$ corresponds to more zonal flow, while a high $S_{ag}$ corresponds to a wavier (i.e., more meridional) flow. As an example, Fig. 1a displays the synoptic setting on August 16th, 2001 when a zonal flow pattern resulted in a low $S_{ag}$ of 1.09. In clear contrast, Fig. 1b shows the synoptic setting on July 11th, 2012 where an $S_{ag}$ of 2.00 accompanied a high-amplitude Omega blocking pattern that was responsible for exceptional melt of the GrIS that eventually extended to over 98% of the ice sheet's surface[4].

We also followed ref. 19 in computing sinuosity as a function of latitude. This was done at daily time steps by first calculating $S_i$ at 10 m increments spanning the Northern Hemisphere, then assigning the value of $S_i$ to the latitude whose zonal-mean geopotential height matched the value of the isohypse, $i$.

The sinuosity index effectively indicates whether the type of wavy circulation that is conducive to persistent, anomalous weather is present over the North Atlantic. To more specifically measure whether this wavy flow produces blocking over Greenland, we employ the GBI, which is simply the latitude-weighted mean Z500 over a domain spanning 60–80°N and 20–80°W[39,40]. Thus, the GBI falls into the broad category of anomaly-based blocking detection methods, as Greenland blocking episodes are often identified as positive GBI anomalies relative to the time mean[3,7,63]—an approach that emphasizes the presence of an anomalous anticyclone as the defining feature of atmospheric blocking. The GBI has been the primary index used to study the relationship between blocking and Greenland surface mass balance[7,8,40] as well as to establish the recent positive trend in summer Greenland blocking[3,7,8]. Furthermore, monthly- and seasonal-mean GBI values are strongly correlated with the frequency of the exceptionally high GBI days that typify Greenland blocking episodes[3].

### Investigating lagged relationships between snow cover and summer atmospheric circulation over Greenland

To investigate whether summer atmospheric circulation over Greenland is influenced by Arctic-amplified changes in high-latitude snow cover, we performed a lagged linear regression of each of the three circulation indices detailed in the previous section against antecedent snow cover extent (SCE). Specifically, we examined Northern Hemisphere, Eurasian, and North American SCE area as derived from the National Oceanic and Atmospheric Administration (NOAA) Climate Data Record (CDR) of Northern Hemisphere SCE by the Rutgers University Global Snow Lab[64]. The CDR provides weekly SCE observations that are primarily produced via manual interpretation of visible satellite imagery[23,65]. These observations are made available on an 88×88 polar stereographic grid spanning the Northern Hemisphere and, given their course spatial resolution, are most suitable for monitoring SCE at continental scales[23]. Previous applications of the SCE CDR that are particularly relevant to this study's objectives include documenting

the rapid retreat of spring snow-cover across the Northern Hemisphere as an indicator of AA[22] and demonstrating the influence of Eurasian SCE variability on summer atmospheric circulation[42].

### Tracing the atmospheric response to snow cover variability

For instances where the lagged regression analysis revealed an apparent link between spring SCE and the ensuing summer atmospheric circulation over Greenland, we looked for evidence of a direct stationary Rossby wave response as a potential explanatory dynamical mechanism underlying the observed statistical relationship. This was accomplished by tracing the stationary wave response using the three-dimensional wave activity flux, where the WAF vectors run parallel to the local group velocity of the Rossby wave packet[46].

Previous studies have applied this formulation of the WAF to demonstrate the influence of both sea ice[28,29] and snow cover variability[33] on atmospheric circulation by displaying the WAF calculated from composite-difference fields representing the change in the state of the atmosphere between years of opposing surface conditions (e.g., low-minus-high SCE years). Here, we follow the general approach of ref. 66 by calculating the WAF from regression coefficients rather than composite differences to provide a more direct representation of the conditions captured in the lagged regression analysis. Specifically, we calculated the WAF from the geostrophic stream function anomaly derived from the coefficients of a linear regression of ERA5 Z300 against inverted monthly SCE area performed at each grid cell of the reanalysis data. Thus, our results represent the atmospheric response to one standardized negative SCE anomaly over the 1979–2022 study period. The circulation indices and Z300 were detrended prior to regression, and the regression coefficients were mapped to a 2.5° grid using first-order conservative remapping and smoothed using a Gaussian filter prior to computing the WAF.

One proposed chain of mechanisms linking summer atmospheric circulation to spring surface conditions is the snow-hydrological effect, whereby diminished spring SCE causes a depletion of soil moisture and, consequently, enhanced surface heating in summer[33,34]. To investigate this pathway, we also perform a grid point linear regression of detrended, monthly ERA5 soil moisture and surface air temperature against detrended monthly SCE. We perform the regression using monthly surface variables for each month spanning the significant lagged correlation between spring SCE and summer atmospheric circulation to illustrate the temporal chain of mechanisms in the snow hydrological effect.

### Statistics

In all cases, the null hypothesis that the slope of each linear regression performed in this analysis was equal to zero was tested using a two-tailed t-test and a significance level of $\alpha = 0.05$. We also employed linear regression to assess the long-term trends in June North American and Eurasian SCE area, as well as JJA $S_{ag}$ and GBI. In each case, we found no evidence of temporal autocorrelation in these monthly and seasonal time series (Supplementary Fig. 6 and Supplementary Table 2). We therefore determined linear regression to be suitable for the slope estimate. For the idealized model experiment, differences between the control and reduced snow simulations were tested using a two-tailed paired t-test and a significance level of $\alpha = 0.05$. All hypothesis tests were performed without multiple testing adjustment and without considering spatial autocorrelation.

### Idealized model experiment

To support the results based on analysis of ERA5 reanalysis presented in the main manuscript, we conducted a controlled idealized experiment using the National Center for Atmospheric Research's Community Earth System Model version 2.2 (CESM2)[67]. CESM2 was run in the Atmospheric Model Intercomparison Project (AMIP) configuration, which includes interactive atmosphere and land-surface components

with prescribed sea surface temperatures (SSTs) and sea-ice. SSTs and sea-ice were prescribed as monthly varying seasonal cycles based on the observed climatology from 2005 to 2015 (i.e., component set: F2010climo)[68]. The atmosphere component was the Community Atmosphere Model version 6 (CAM6)[69] and the land-surface component was the Community Land Model version 5 (CLM5)[70], both of which had corresponding horizontal resolutions of 0.9° (latitude) by 1.25° (longitude). We ran this configuration for ten consecutive years (control simulation) and saved restart files for May 1st of each year. We then modified the land-surface restart files by reducing the snow cover over North America to zero and reran 3-month simulations (May 1st to July 31st) for each of the 10 years (reduced snow simulation). This approach tested the impact of snow cover while maintaining inter-annual variability. Our analysis, presented in the supplementary information and discussed in the section "Preliminary GCM results", assessed the differences between the reduced snow and controlled simulations to quantify the response to low May North American snow cover. Supplementary Fig. 3 presents the atmospheric response and Supplementary Fig. 4 demonstrates that removal of May snow cover resulted in the anticipated impact on surface conditions over North America—i.e., reduced spring snow cover was followed by depleted soil moisture, enhanced sensible heat flux, and warmer surface temperatures.

## Data availability

ERA5 reanalysis obtained using Copernicus Climate Change Service at https://cds.climate.copernicus.eu/cdsapp#!/home. Snow cover extent area timeseries available through Rutgers University Global Snow Lab at https://climate.rutgers.edu/snowcover. The NOAA SCE CDR is available at https://www.ncei.noaa.gov/products/climate-data-records/snow-cover-extent. NAO index values are available at https://climatedataguide.ucar.edu/climate-data/hurrell-north-atlantic-oscillation-nao-index-pc-based. The CESM experiment output generated in this study have been deposited in the Zenodo repository accessible at https://doi.org/10.5281/zenodo.7974703.

## Code availability

The CESM code is maintained by NCAR and available via GitHub (https://github.com/ESCOMP/CESM/tree/release-cesm2.2.0). The authors will make available any code used for data analysis or figure generation upon request.

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

## Acknowledgements

J.P. and T.M. are supported by NSF Arctic Systems Science award number 1900324. J.C. is supported by the US NSF grant PLR-1901352 and OPP-2115072. M.T. is supported by NSF OPP-1713072, OPP-1901603, and Heising-Simons Foundation HSFOUND 2019-1160. G.J.K. acknowledges support from the U.S. Department of Energy (DOE) Regional and Global Model Analysis (RGMA) Program (DE-SC0021209). Computing resources for the Community Earth System Model (CESM) simulations were provided on the Cheyenne supercomputer by the Computational and Information Systems Laboratory (CISL) at the National Center for

Atmospheric Research (NCAR). Development of CESM is led by the NCAR, which is supported primarily by the National Science Foundation under Cooperative Agreement No. 1852977.

## Author contributions

J.R.P. performed the formal analysis, created the visualizations, and wrote the manuscript. J.R.P., T.L.M., J.C., and M.T. conceptualized the study. L.J.W. contributed to the methodology and calculation of atmospheric circulation indices. T.L.M., J.C., M.T., J.A.K., and G.J.K. contributed to the methodology and interpretation of the results. G.J.K. completed the idealized model simulations.

## Competing interests

The authors declare no competing interests.
