## [Peer Review File · Nature Communications]

Summer Atmospheric Circulation Over Greenland in Response to Arctic Amplification and Diminished Spring Snow CoverREVIEWER COMMENTS

Reviewer #1 (Remarks to the Author):

In this manuscript, the authors study atmospheric circulation with the goal of ascertaining if there have been changes to blocking conditions affecting Greenland. The authors conjecture recent decades have experienced increased waviness of the jet and that this is linked to Greenland blocking. They further conjecture that changes in snow/ice cover are causing the increased waviness.

In general, I find that the description of the statistical methods lacking in precision. I expound below.

In Section 2.1 the authors study sinuosity of isohypes. Specifically, they study S_{ag} which I believe is an annual summary of sinuosity aggregated over several isohypes and also over the summer season of each year. The authors could make this more clear by saying $S_i(t)$ is the sinuosity of isohypse i and day t . Then $S_{ag}(t)$ would be aggregated across i (mean or sum?) and $S_{ag} = \text{Sum}_{t}^{T} S_{ag}(t)$ where T is the number of days in the season. Figure 2 shows S_{ag} for a 40 year period. The authors' definition of sinuosity seems reasonable to me.

Presumably because the annual values are highly variable, in Figure 2 the authors show a running 5 year mean to better illustrate trends. In the text, immediately after describing the 5 year running mean, the authors state (l. 130-133) they estimate the increase in GBI to be 0.42 SD per decade. Given the authors were just discussing the running average, it gives the impression that this estimate was calculated from the running average, and not the annual data. If so, this is incorrect. A slope estimate should be based on the annual data and it should not assume the errors are independent. It looks as if the annual data are negatively correlated in time, and a simple auto-covariance error structure (AR1 might be enough?) should be used for slope estimation. The authors should also give a slope estimate for S_{ag} . Importantly, the auto-correlation will affect the significance of the test. P-values should be reported (not just $\alpha = 0.05$), or 95% confidence intervals would be even better. It is also unclear whether the reported correlation (l 133) is on the annual data or the running mean.

The paragraph on line 134 talks about a different blocking index PH03. I don't think this adds to the paper and including it in the graphs makes them more complicated. I suggest just briefly discussing how PH03 and GBI show poor correspondance and removing it from the remainder of the paper. Maybe include PH03 results in the supplementary materials.

The authors go on to show correlation between daily values (in my notation, $S_{ag}(t)$ and $GBI(t)$). The authors have shown correlation and not causation, and they should be careful in their language to reflect this. Particularly l 160 says wavy circulation "fosters" blocking.

The authors then go on to study sinuosity as a function of latitude, and this discussion centers on Fig 4. I think the analysis that lies behind Figure 4 needs to be changed. As I understand it, the authors fit a linear regression for data beginning in the year given along the x axis and the end of the data record. If true, then the authors are comparing regressions to time periods of different length and data being used in each regression is not independent (b/c the same data is being used multiple times). The stippling showing shown in the figure has all kinds of multiple comparison and data dependence issues. I suggest the authors discard this analysis and break the time period into non-overlapping time periods of 10 or 20 years.

In Section 2.2 the authors seek to relate snow cover to atmospheric circulation. As above, is the slope estimate in line 189 should be based on the annual data not the 5 year running average. And the significance reported should account for temporal autocorrelation. Correlation (l 194) should be on annual data. P-values for tests with null hypothesis of zero correlation should be reported.

I guess I am ok with figure 5. At a minimum, it should be stated that hypothesis tests were conducted with *individual* $\alpha = 0.05$, that is, with no multiple testing adjustment.

I'm afraid I don't understand Figure 6. The caption reads: "Anomalies of July Z300 regressed against inverted May North American SCE (contours: solid, positive; dashed, negative)". I don't know what this means. Do the contours represent the change in response to a one unit increase in SCE? Was a regression performed for each grid cell to achieve the contours? What is the contour interval? What is "associated" WAF? When you say anomalies, what is the standard from which the anomaly is defined?

It should be mentioned that the hypothesis tests associated with the stippling in Figures 7 and 8 are done individually without multiple testing adjustment or accounting for spatial correlation (which would be more difficult!)

Figure 8: what does it mean for an anomaly to be "obtained from a linear regression"? I still don't know what the standard is used to define the anomaly.

Minor points:

I 63: weather extremes here is too vague. What type of extreme?

I 95: Will all readers of this article understand what "ridging" is?

I 134: I wouldn't say PH03 results "stand in contrast" as they have similar trend as GBI, but just not as strong, right?

I 238/: Figure 6a not 8a, right?

Reviewer #2 (Remarks to the Author):

The authors investigated the dynamical influence of low spring North American snow cover on stronger Greenland blocking. Another prominent mechanism would be the highlatitude warming related to AA allowing for wavier jet and hence frequent or stronger prevalent Greenland blocking. With respect to the latter mechanism, it is less than novel since existing studies have already presented evidence to demonstrate this link, as also indicated by the authors. The highlight of this study would be the dynamical response to spring snow cover retreat. The topic is interesting and suitable for the journal. The observed (and modelling) evidence presented by the authors to explain the mechanisms, however, appears to be insufficient and missing. I would recommend the paper undergoes major revisions to conduct appropriate analysis and numerical simulations.

1. The authors use ERA5 reanalysis to investigate the impact of North American SCE retreat on stronger Greenland blocking. Although the physical mechanisms unveiled by the observational dataset seem reasonable, some details are worthy of further discussion (see comment 4). I would suggest the authors conduct AGCM numerical simulations to demonstrate the blocking response to surface forcings further. Such a comparison of observational study and modelling study would be more convincing.

2. The main body of this paper links the stronger Greenland blocking to the North American snow retreat in May, but in Fig. 2 the authors analyzed the NH SCE change in June. Could the authors show the temporal change of SCE in both North America and Eurasia to look for closer connections between SCE and GBI? Another issue would be the interdecadal change of GBI, which underwent abrupt climate change around 2010, differing from the SCE change since 2010. The continuous decline of snow cover after 2010 corresponded to a trend of weaker blocking. I am curious about the physical connection and mechanism underlying the reversed relationship.

3. Fig. 4 shows the long-term trends of variables. Does the atmospheric circulation anomaly in association with retreating SCE show a similar feature as the trend? Additional analysis of sinuosity and zonal wind anomalies in response to SCE loss would be valuable to contrast with the climate trends. The authors also emphasize the AA influence on Greenland blocking, but the analysis of how SCE impacts AA is omitted.

4. The strongest origin of upper tropospheric response and WAF is located around Hudson Bay, where there hardly seen any SCE signals and SAT anomalies. Could you please provide an explanation for the reason why such poor surface condition anomaly can induce the strongest WAF? The authors identify the eastern North America warm center as the wave source of Rossby waves that affect the Greenland blocking, however, this center emerged suddenly and appeared to be independent of (or irrelevant to) the snow cover loss over the American continent, albeit significant correlation with the May SCE.
5. From Fig. 4 we can see another source of July blocking variability, i.e. the June Eurasian snow cover. I suggest the authors conduct a supplementary analysis of the Eurasian snow cover impact on Greenland blocking. I can understand that the revision time left for the authors would not be sufficient. At least a section of discussion on the Eurasia-Greenland linkage is necessary.

Minor Comments:

1. Line 238: Fig. 8a should be Fig. 6a
2. The current definition of blocking treated blocking as a climate phenomenon, as the area-average of geopotential height is utilized. But in common sense, blocking is representative of a short-time synoptic event, and the temporal duration of a blocking event is usually considered in its definition.
3. Could the authors show the interdecadal change of geopotential height pattern related to the Greenland blocking?

Reviewer #3 (Remarks to the Author):

Review of "summer atmospheric circulation over Greenland in response to Arctic Amplification and Diminished snow cover" by Jonathon Preece et al.

The manuscript discusses the mechanism causing the observed increase of summer Greenland Blocking, pointing to the role of the Arctic Amplification and specifically to the retreat of snow cover over the Eastern North American continent. Such anomaly, more evident in recent years, produces a surface warming in spring which triggers a wave activity flux able to increase the frequency of summer Greenland blocking. The analysis is carried out making use of different indices, from blocking to sinuosity, and it is both well-presented and detailed. Overall I enjoyed the manuscript and I believe it can be a fruitful contribution to the topic, since to my knowledge a reasonable explanation for the increased Greenland Blocking trend is lacking: however, I have some concerns about the robustness of the discussed trend – which might be more of a variability rather than a trend – and consequently the role of the Arctic Amplification in driving this changes.

I thus suggest the manuscript to be accepted following major revisions.

Main Comments

- My main concern regards the time window of the analysis. North Atlantic variability, and especially Greenland blocking, are known for having a large interannual variability: a single year with high blocking frequency can be followed by a year without any blocking. I would encourage the authors to extend the analysis to pre-satellite data – ERA5 is now definitive up 1959 if I am correct – considering that North Atlantic and North America are widely covered by pre-satellite observations. Also, I would strongly suggest to include summer 2020 and 2021 (and perhaps 2022) into the analysis, since as far as I know they have been characterized by low Greenland Blocking frequency, which I imagine can influence the final results.
- In Figure 4 a reduction of the trend of sinuosity in recent years is presented. This weakening of the trend puzzles me a bit – also in combination to the above comment. This is quite intriguing to me because while the changes in sinuosity are reflected by the change in the zonal winds, as shown by Figure 4c (and this make sense: due to the presence of GB the jet shifts southward, so that weakens at high latitude and increase at lower latitude as shown), a similar trend change it is not seen in lower tropospheric temperature. If is the Arctic Amplification that is driving the increase of GB, why the trend

of high latitude temperature (around 70-80N) is still increasing in more recent years while sinuosity trend is weakening? Actually high latitude sinuosity is getting almost to zero. I think that the authors should clarify this point, since as it is currently presented the text is a bit biased suggesting that the AA is responsible for most of the above discussed changes. Personally, I am not in the position to exclude (from what I see) that the observed variability is due only to internal variability.

- It would be interesting to see – also considering the analysis carried out in Figure 5 – in which month the sinuosity/GBI trends are significant and in which not. Indeed, it is shown that there is a global trend in JJA. However, the analysis presented later on seems to support that the direct influence of spring SCE on summer sinuosity is significant mostly in July (Figure 5bf). Consequently, the results suggest that the trend of JJA ascribable mainly to the trend in July. Is this confirmed by the single month trend analysis?

- I am not convinced by the explanation regarding the discrepancies between GBI and PH03. Indeed, blocking over Greenland is characterized by a very strong overturning, and such indices have been shown to capture very well the nature of the cyclonic wave breaking up there. My speculation is that the comparison you are doing is biased by the choice of using the original 1D index from Pelly and Hoskins (2003), rather than a 2D extensions as the one developed by Berrisford et al. (2007) and used by Woollings et al. (2008) or Masato et al. (2011). Indeed, using the single latitude based on EKE can be a very good metric, but it partially misses high latitude blocking as GB, since it passes too south of Greenland (see PH03 Figure 3). I encourage the authors to re-do the computation defining a 2D blocking index and then using a box-average over Greenland to define the presence of blocking on seasonal basis: I am quite convinced that this correlation will be largely increased. If they do not want to use theta-on-PV, a similar result should come out if you use a reversal index based on Z500 as the Scherrer et al (2004).

Minor Comments

L47: Please consider that NAO and Greenland blocking are simply two facets of the same dice, as shown by Woollings et al. (2008)

L56: A consistent analysis of GB in both models and reanalysis can be found also in Davini and D'Andrea (2020), which showed that none of the CMIP5/CMIP6 models is able to replicate the trend.

L120: is the standard deviation that increases or the average value of the timeseries that increase by 0.33 standard deviation? The standard deviation is computed over the entire period? Please clarify.

L121: which percentile you are discussing here? It is a bit unclear how do you measure your statistics here: at first, I assumed you have a single average value of Sag per season, which provide you a trend, but this does not seem to be the case. Do you mean that you compute percentiles on the each year daily Sag values and then assess their trend? If this is the case, please clarify the text since as it is written it is not obvious.

L126: please use a reference also for the GBI

L134: are correlation in Figure 2b detrended? Please clarify that the trend has an impact on the correlations.

L160: Do you have any speculation while the fit tend to bend in favor of higher GBI for a lower sinuosity? Also, please keep in mind that NAO-GB relationship are not strictly linear, as shown by Woollings et al. (2010)

L170: I wonder how you can compute sinuosity at 80N: if you have a isolines which crosses the pole, how this is evaluated? Is this Sag still?

L194-196: this comment on the trend-induced correlation should go when discussing Figure 2.

L224: I wonder what the correlation in Figure 5c,f means: a positive correlation between February SCE over North America and August GBI seems a bit counterintuitive. Since it is a quite surprising result, it would be nice if the authors can say something about it.

L238: Figure 6a, not 8a

L239: “the vertical (colors) and horizontal (arrows) components of the associated WAF and its horizontal divergence (stippling)”

L280: Why do the authors prefer to use snow cover duration instead of a simple snow cover fraction? Or are actually the same thing?

L281: are these correlations detrended?

L289: it would be very informative to have also sensible heat flux here, since it is the driving mechanism that could sustain the wave activity flux shown in Figure 6. ERA5 data are available in this sense.

L289: what is defined as “spring” SCE? Is this May North America SCE? Please clarify (also in Figure 8 caption).

L316-326: Honestly, I am not getting the added value of Figure 9. I would suggest that the authors remove it.

L338: as discussed in the main comments, I think this claim is a bit too strong. As long as I see from Figure 5 AA is still increasing in recent years, while the GB trend is weakening.

L340-341: Actually this is not completely true: wave breaking can be cyclonic, and thus oriented over Greenland, or anticyclonic, and thus producing the most common European blocking. There are not so much different alternative, so that cyclonic wave breaking almost always occur over Greenland, as shown by multiple studies on atmospheric blocking (the already referenced Woollings et al. 2018 or Davini and D’Andrea 2020 are good examples). Please rephrase.

L366-369: I am not understanding why there is a mention to Omega block here. The geopotential height anomalies in Figure 5 show a reversal of the flow with negative anomalies over North Atlantic, which could remind even a Rex block. I would remove the reference to the Omega block here which seem to ma bit forced.

L370: “retreating North American SCE”.

Figure 2b: rather than using the colors for highlighting the magnitude of the correlation, it would be more useful to use it to mark significant/non-significant correlations.

Figure 5. Please use lettering for panels and not row/columns.

REVIEWER COMMENTS

Reviewer #1 (Remarks to the Author):

In this manuscript, the authors study atmospheric circulation with the goal of ascertaining if there have been changes to blocking conditions affecting Greenland. The authors conjecture recent decades have experienced increased waviness of the jet and that this is linked to Greenland blocking. They further conjecture that changes in snow/ice cover are causing the increased waviness.

In general, I find that the description of the statistical methods lacking in precision. I expound below.

In Section 2.1 the authors study sinuosity of isohypes. Specifically, they study $S_{\{ag\}}$ which I believe is an annual summary of sinuosity aggregated over several isohypses and also over the summer season of each year. The authors could make this more clear by saying $S_i(t)$ is the sinuosity of isohypse i and day t . Then $S_{\{ag\}}(t)$ would be aggregated across i (mean or sum?) and $S_{\{ag\}} = \text{Sum}_{\{t\}}^T S_{\{ag\}}(t)$ where T is the number of days in the season. Figure 2 shows $S_{\{ag\}}$ for a 40 year period. The authors' definition of sinuosity seems reasonable to me.

We have made the suggested changes to the notation.

Presumably because the annual values are highly variable, in Figure 2 the authors show a running 5 year mean to better illustrate trends. In the text, immediately after describing the 5 year running mean, the authors state (l. 130-133) they estimate the increase in GBI to be 0.42 SD per decade. Given the authors were just discussing the running average, it gives the impression that this estimate was calculated from the running average, and not the annual data. If so, this is incorrect. A slope estimate should be based on the annual data and it should not assume the errors are independent. It looks as if the annual data are negatively correlated in time, and a simple auto-covariance error structure (AR1 might be enough?) should be used for slope estimation. The authors should also give a slope estimate for $S_{\{ag\}}$. Importantly, the auto-correlation will affect the significance of the test. P-values should be reported (not just $\alpha = 0.05$), or 95% confidence intervals would be even better. It is also unclear whether the reported correlation (l 133) is on the annual data or the running mean.

We did estimate the slope on the annual time series and not the 5-year running mean. Thank you to the reviewer for pointing out how the original writing could be misleading. We have removed the reference to the running mean and explicitly stated that the linear model was applied to the annual data to avoid any confusion.

Regarding the need for an autoregressive model, we examined the residuals of the linear models for each of the time series in Fig. 2 via their autocorrelation functions and a Durbin-Watson test (see the newly added Supplementary Figures 6-7 and Supplementary Table 2). This revealed no discernable autocorrelation for either time series.

The slope estimate for $S_{\{ag\}}$ is noted earlier in the paragraph. We have extended the slope estimates from the linear model through 2022 and included the p-values and 95% confidence intervals.

The paragraph on line 134 talks about a different blocking index PH03. I don't think this adds to the paper and including it in the graphs makes them more complicated. I suggest just briefly discussing how

PH03 and GBI show poor correspondance and removing it from the remainder of the paper. Maybe include PH03 results in the supplementary materials.

Thank you to the reviewer for the insightful comment. We agree that including the results from PH03 is not critical to the presentation of our key findings. Furthermore, and as noted in our response to Reviewer 3, the relatively poor performance of flow-reversal-based metrics such as PH03 in capturing the recent positive trend in summer Greenland blocking has been documented elsewhere—a point that does not need to be relitigated here. Considering this, we have elected to exclude the PH03 results from the manuscript and focus on the GBI, which has been the primary metric used to document the increase in Greenland blocking.

The authors go on to show correlation between daily values (in my notation, $S_{ag}(t)$ and $GBI(t)$). The authors have shown correlation and not causation, and they should be careful in their language to reflect this. Particularly l 160 says wavy circulation "fosters" blocking.

This is a very important distinction. We have softened our language as follows so that we do not overstate the meaning of our results:

"...signaling a strong tendency for wavy circulation over the North Atlantic to coincide with anticyclonic conditions over Greenland."

The authors then go on to study sinuosity as a function of latitude, and this discussion centers on Fig 4. I think the analysis that lies behind Figure 4 needs to be changed. As I understand it, the authors fit a linear regression for data beginning in the year given along the x axis and the end of the data record. If true, then the authors are comparing regressions to time periods of different length and data being used in each regression is not independent (b/c the same data is being used multiple times). The stippling showing shown in the figure has all kinds of multiple comparison and data dependence issues. I suggest the authors discard this analysis and break the time period into non-overlapping time periods of 10 or 20 years.

We believe that the analysis (which is the same as that employed in the Vavrus et al., 2017 Journal of Climate article) is uniquely suited for illustrating both the spatial and temporal signature of the trends. So, we would like to keep the general form of the analysis, if possible. However, we appreciate the reviewer's concerns and see that the presentation of Fig. 4 needs adjustment.

Regarding the issue of comparing regressions of varying lengths, we have switched from using an expanding window regression to a true rolling regression with a constant window size of 25 years. While this does involve using the same data multiple times, we believe that it also makes a stronger case that the trends are robust given that they show that the results yield low p-values spanning a range of endpoints.

To address the issues of multiple testing, we have elected to remove the stippling and present a companion set of panels which display the p-values for each regression rather than using a specific significance threshold. Additionally, and in line with the reviewer's comments elsewhere, we have noted that the regressions for each latitude and time window were conducted independently with no multiple testing adjustment.

In Section 2.2 the authors seek to relate snow cover to atmospheric circulation. As above, is the slope estimate in line 189 should be based on the annual data not the 5 year running average. And the significance reported should account for temporal autocorrelation. Correlation (l 194) should be on annual data. P-values for tests with null hypothesis of zero correlation should be reported.

Here again the autocorrelation function and Durbin-Watson test did not suggest autocorrelation in the snow cover time series (Supplementary Fig. 6 and Supplementary Table 2). The slope estimates were based on the annual data and we have clarified this in the writing. We have also updated the slope estimates to run through 2022 and included the corresponding p-values and 95% confidence intervals.

I guess I am ok with figure 5. At a minimum, it should be stated that hypothesis tests were conducted with *individual* $\alpha = 0.05$, that is, with no multiple testing adjustment.

We have added a statement to the caption of Fig. 5 to indicate that there was no multiple testing adjustment. Thank you for helping to describe the statistical results more precisely.

I'm afraid I don't understand Figure 6. The caption reads: "Anomalies of July Z300 regressed against inverted May North American SCE (contours: solid, positive; dashed, negative)". I don't know what this means. Do the contours represent the change in response to a one unit increase in SCE? Was a regression performed for each grid cell to achieve the contours? What is the contour interval? What is "associated" WAF? When you say anomalies, what is the standard from which the anomaly is defined?

Thank you for highlighting this ambiguity in the presentation of our results. We have made edits to the figure caption as well as section 4.3 to clarify that the wave activity flux was calculated from the coefficients of a grid-point linear regression of 300 hPa geopotential height against standardized 1979-2022 SCE area. We have also clarified that Figure 6 represents the atmospheric response to one negative standardized anomaly of May North American SCE area. The contour interval is now noted in the caption for Fig. 6a.

It should be mentioned that the hypothesis tests associated with the stippling in Figures 7 and 8 are done individually without multiple testing adjustment or accounting for spatial correlation (which would be more difficult!)

We have added text to each figure caption and at the end of section 4.3 to acknowledge this fact.

Figure 8: what does it mean for an anomaly to be "obtained from a linear regression"? I still don't know what the standard is used to define the anomaly.

We have edited the figure caption to clarify that the shading shows the linear regression coefficients relating the eddy-geopotential height to North American SCE area.

Minor points:

l 63: weather extremes here is too vague. What type of extreme?

This sentence now clarifies the type of extreme.

I 95: Will all readers of this article understand what "ridging" is?

Thank you for raising this point. We took this as an opportunity to introduce the association between ridging and high pressure.

I 134: I wouldn't say PH03 results "stand in contrast" as they have similar trend as GBI, but just not as strong, right?

As noted above, we have elected to exclude the PH03 results.

I 238/: Figure 6a not 8a, right?

Yes! Thank you for catching this oversight. It is now correct.

Reviewer #2 (Remarks to the Author):

The authors investigated the dynamical influence of low spring North American snow cover on stronger Greenland blocking. Another prominent mechanism would be the highlatitude warming related to AA allowing for wavier jet and hence frequent or stronger prevalent Greenland blocking. With respect to the latter mechanism, it is less than novel since existing studies have already presented evidence to demonstrate this link, as also indicated by the authors. The highlight of this study would be the dynamical response to spring snow cover retreat. The topic is interesting and suitable for the journal. The observed (and modelling) evidence presented by the authors to explain the mechanisms, however, appears to be insufficient and missing. I would recommend the paper undergoes major revisions to conduct appropriate analysis and numerical simulations.

1. The authors use ERA5 reanalysis to investigate the impact of North American SCE retreat on stronger Greenland blocking. Although the physical mechanisms unveiled by the observational dataset seem reasonable, some details are worthy of further discussion (see comment 4). I would suggest the authors conduct AGCM numerical simulations to demonstrate the blocking response to surface forcings further. Such a comparison of observational study and modelling study would be more convincing.

In response to the reviewer's comment, we have completed some preliminary idealized modeling using CESM2. We detail the experimental setup in section 4.4, present the results in Supplementary Fig. 3 and 4, and discuss them in section 2.3.

We found that, when imposing reduced May North American snow cover in CESM2, the model reproduced the observed response to low spring SCE remarkably well. Both the anomalous anticyclone over Baffin Bay and the WAF from Eastern Canada toward Greenland are present in the model output. The modeled response does occur a month earlier than what we found in the reanalysis data; however, as noted in section 2.3, this may be due to the abrupt removal of snow cover in the model experiment, and we plan to explore this further in future work. Given the time constraints of the review process, we feel that a full analysis of the representation of this snow-atmosphere coupling in GCMs would be most effectively completed as part of future work, and we have indicated that we plan to do so in the closing statement of our discussion section.

We very much appreciate the reviewer's comment and believe that, when considered in conjunction with our observational analysis, the preliminary modeling results that have followed from their review clearly

provide more compelling evidence that snow-atmosphere coupling has played a role in the increase in summer Greenland blocking.

2. The main body of this paper links the stronger Greenland blocking to the North American snow retreat in May, but in Fig. 2 the authors analyzed the NH SCE change in June. Could the authors show the temporal change of SCE in both North America and Eurasia to look for closer connections between SCE and GBI? Another issue would be the interdecadal change of GBI, which underwent abrupt climate change around 2010, differing from the SCE change since 2010. The continuous decline of snow cover after 2010 corresponded to a trend of weaker blocking. I am curious about the physical connection and mechanism underlying the reversed relationship.

We have updated Fig. 2a to show the time series of North America and Eurasia SCE area separately and Fig 2b so that each snow cover region is represented in the correlation matrix. We have also noted snow cover trend in each region at the start of section 2.2.

We disagree with the reviewer's characterization of the blocking and snow cover trends post 2010. The declining trend in N. Hemisphere snow cover levels off and undergoes a slight reversal after 2010, similar to what is observed for both atmospheric circulation indices. While there is a decline in GBI and sinuosity after 2010, the respective 5-year running means remain higher than at any point prior the repaid decline in snow cover around the turn of the century. We believe this mirrored behavior in the snow cover and atmospheric circulation indices is consistent with our results showing a relationship between low N. American spring SCE and anomalous ridging over Greenland.

3. Fig. 4 shows the long-term trends of variables. Does the atmospheric circulation anomaly in association with retreating SCE show a similar feature as the trend? Additional analysis of sinuosity and zonal wind anomalies in response to SCE loss would be valuable to contrast with the climate trends. The authors also emphasize the AA influence on Greenland blocking, but the analysis of how SCE impacts AA is omitted.

As is shown in the newly added Supplementary Fig. 1, the anomaly pattern in response to low snow cover does indeed strengthen later in the study period, consistent with the long-term trends in the GBI and sinuosity.

We do directly examine the relationship between snow cover and sinuosity in Fig. 5. While we agree that it would be interesting to look directly at the relationship between snow cover and AA, the objective of this paper was not to quantify the contribution of the snow cover loss to disproportionate warming at high latitudes. Rather, our objective was to look at AA and snow cover loss each as potential contributors to the increase in Greenland blocking. Based on our results we propose that AA may contribute to the increase in blocking by weakening the zonal flow, thereby producing a wavier circulation. We also propose that, under this altered background state, declining N. American snow cover has encouraged ridging over Greenland through a stationary wave response. Neither of these mechanisms are dependent upon snow cover being a primary cause of summer AA.

However, in section 2.2 we do simply raise the possibility that retreating snow cover may also be indirectly linked to increase in Greenland blocking through its contribution to Arctic amplification, which in turn weakens the zonal flow. Previous work has frequently pointed to the likely contribution of snow cover loss to AA and we have added the appropriate citations to refer to this work in support of our

statement in section 2.2. We have also edited the opening sentences of the second-to-last paragraph in section 3 to be more firmly based in our results.

4. The strongest origin of upper tropospheric response and WAF is located around Hudson Bay, where there hardly seen any SCE signals and SAT anomalies. Could you please provide an explanation for the reason why such poor surface condition anomaly can induce the strongest WAF? The authors identify the eastern North America warm center as the wave source of Rossby waves that affect the Greenland blocking, however, this center emerged suddenly and appeared to be independent of (or irrelevant to) the snow cover loss over the American continent, albeit significant correlation with the May SCE.

We have made edits to sections 2.2 and 2.4 to clarify our reasoning for the stationary wave response. In section 2.2 we note that Fig 6c shows that years of low North American snow cover are associated with an anomalous baroclinic structure in the atmospheric column over Eastern Canada. The westward tilt of the eddy geopotential heights in this region supports the upward propagation of Rossby waves between ~54 and 62° N, which then turn northeastward. As the reviewer mentions, this horizontal component becomes apparent over Hudson Bay before then converging over Greenland. We note that this northeast path of the WAF is nicely captured in the model experiment (Supplementary Fig. 3b).

In section 2.4 we go on to identify the warm anomaly over Eastern Canada as the reason for the baroclinic structure in the overlying atmosphere. Section 2.4 also details how this warm anomaly is linked to snow cover. The sudden emergence of the warm anomaly is a byproduct of the delayed influence of snow cover on soil moisture (Fig 8d and 8g). Depleted soil moisture increases heating of the atmosphere over Eastern Canada, which induces the baroclinic response and accompanying upward WAF over the same region (Fig 6c).

5. From Fig. 4 we can see another source of July blocking variability, i.e. the June Eurasian snow cover. I suggest the authors conduct a supplementary analysis of the Eurasian snow cover impact on Greenland blocking. I can understand that the revision time left for the authors would not be sufficient. At least a section of discussion on the Eurasia-Greenland linkage is necessary.

Thank you for your understanding of the time constraints around completing the revisions. As suggested by the reviewer, we have expanded our discussion of the apparent relationship with Eurasian SCE in section 2.2 to include some discussion of Matsumura et al. (2011), which establishes a stationary wave response to low spring Eurasian snow cover.

Minor Comments:

1. Line 238: Fig. 8a should be Fig. 6a

The figure reference on Line 238 has been corrected.

2. The current definition of blocking treated blocking as a climate phenomenon, as the area-average of geopotential height is utilized. But in common sense, blocking is representative of a short-time synoptic event, and the temporal duration of a blocking event is usually considered in its definition.

Thank you for raising this point regarding the timescale of blocking. We selected the GBI because it has been the primary metric used to establish the recent positive trend in summer Greenland blocking, and monthly and seasonal means of the index are strongly correlated with the frequency of high GBI days

(see Hanna et al., 2018; Hofer et al., 2017; Liu et al, 2016; McLeod and Mote, 2016; Tedesco and Fettweis, 2022). We edited section 4.1 to note the correspondence between the frequency of exceptional GBI days and monthly-to-seasonal means of the index.

Hanna, E., Hall, R. J., Cropper, T. E., Ballinger, T. J., Wake, L., Mote, T., & Cappelen, J. (2018). Greenland blocking index daily series 1851–2015: Analysis of changes in extremes and links with North Atlantic and UK climate variability and change. *International Journal of Climatology*, 38(9), 3546–3564. <https://doi.org/10.1002/joc.5516>

Hofer, S., Tedstone, A. J., Fettweis, X., & Bamber, J. L. (2017). Decreasing cloud cover drives the recent mass loss on the Greenland Ice Sheet. *Science Advances*, 3(6), e1700584. <https://doi.org/10.1126/sciadv.1700584>

Liu, J., Chen, Z., Francis, J., Song, M., Mote, T., & Hu, Y. (2016). Has Arctic Sea Ice Loss Contributed to Increased Surface Melting of the Greenland Ice Sheet? *Journal of Climate*, 29(9), 3373–3386. <https://doi.org/10.1175/JCLI-D-15-0391.1>

McLeod, J. T., & Mote, T. L. (2016). Linking interannual variability in extreme Greenland blocking episodes to the recent increase in summer melting across the Greenland ice sheet: EXTREME GREENLAND BLOCKING AND SUMMER MELTING ACROSS THE GREENLAND ICE SHEET. *International Journal of Climatology*, 36(3), 1484–1499. <https://doi.org/10.1002/joc.4440>

Tedesco, M., & Fettweis, X. (2020). Unprecedented atmospheric conditions (1948–2019) drive the 2019 exceptional melting season over the Greenland ice sheet. *Cryosphere*, 14, 1209–1223. <https://doi.org/10.5194/tc-14-1209-2020>

3. Could the authors show the interdecadal change of geopotential height pattern related to the Greenland blocking?

It is our understanding that the reviewer may be concerned with how the geopotential height anomaly associated with low North American SCE might change depending on the time period that is examined. To explore this, we repeated a grid point correlation between July Z300 and May North American SCE multiple 20-yr time windows throughout our study period. Those results are now presented as Supplementary Fig. 1 and they demonstrate that the pattern in Fig. 5 is robust response to low snow cover throughout the study period.

Reviewer #3 (Remarks to the Author):

Review of “summer atmospheric circulation over Greenland in response to Arctic Amplification and Diminished snow cover” by Jonathon Preece et al.

The manuscript discusses the mechanism causing the observed increase of summer Greenland Blocking, pointing to the role of the Arctic Amplification and specifically to the retreat of snow cover over the Eastern North American continent. Such anomaly, more evident in recent years, produces a surface warming in spring which triggers a wave activity flux able to increase the frequency of summer Greenland blocking. The analysis is carried out making use of different indices, from blocking to sinuosity, and it is both well-presented and detailed. Overall I enjoyed the manuscript and I believe it can be a fruitful contribution to the topic, since to my knowledge a reasonable explanation for the

increased Greenland Blocking trend is lacking: however, I have some concerns about the robustness of the discussed trend – which might be more of a variability rather than a trend – and consequently the role of the Arctic Amplification in driving this changes.

I thus suggest the manuscript to be accepted following major revisions.

Main Comments

- My main concern regards the time window of the analysis. North Atlantic variability, and especially Greenland blocking, are known for having a large interannual variability: a single year with high blocking frequency can be followed by a year without any blocking. I would encourage the authors to extend the analysis to pre-satellite data – ERA5 is now definitive up 1959 if I am correct – considering that North Atlantic and North America are widely covered by pre-satellite observations. Also, I would strongly suggest to include summer 2020 and 2021 (and perhaps 2022) into the analysis, since as far as I know they have been characterized by low Greenland Blocking frequency, which I imagine can influence the final results.

The reviewer raises an important point about ending our study period on a year strong summer Greenland blocking. To address this, we have extended our analysis through 2022 as suggested by the reviewer. However, we have elected to continue to limit our study period to the satellite era. While the North Atlantic and North America are relatively well observed, there is still substantial improvement in the quality and coverage of assimilated observations from ~1980 onward. We also note that there are known issues with the initialization of soil moisture in ERA5 prior to this point, which is a key aspect of our analysis. We have made edits to section 4.1 to detail this rationale.

We found our revised results to be robust to the extension of our study period through 2022. Additionally, the strong agreement between our reanalysis-based results and the results of the idealized GCM simulations we performed at the request of Reviewer 2 (now included as Supplementary Fig. 3) increases confidence in our findings despite the relatively short time window of the study period.

- In Figure 4 a reduction of the trend of sinuosity in recent years is presented. This weakening of the trend puzzles me a bit – also in combination to the above comment. This is quite intriguing to me because while the changes in sinuosity are reflected by the change in the zonal winds, as shown by Figure 4c (and this make sense: due to the presence of GB the jet shifts southward, so that weakens at high latitude and increase at lower latitude as shown), a similar trend change it is not seen in lower tropospheric temperature. If is the Arctic Amplification that is driving the increase of GB, why the trend of high latitude temperature (around 70-80N) is still increasing in more recent years while sinuosity trend is weakening? Actually high latitude sinuosity is getting almost to zero. I think that the authors should clarify this point, since as it is currently presented the text is a bit biased suggesting that the AA is responsible for most of the above discussed changes. Personally, I am not in the position to exclude (from what I see) that the observed variability is due only to internal variability.

We have altered the methodology behind Fig.4 in response to the comments of Reviewer 1. Our interpretation of the results is that there is very strong spatial and temporal correspondence between the trends depicted in the figure. The increase in sinuosity (Fig 4a), weakening of the meridional temperature gradient (Fig. 4b), and reduction in the zonal wind (Fig 4c) are all centered around 60 N and the trend in all three variables weakens for 25-year moving windows centered around 2008 and later. This is consistent with the response to a weakened meridional temperature gradient hypothesized in Francis and Vavrus (2012); however, we appreciate the reviewer's concern that we do not overstep our results.

We have adjusted our language in section 3 to state that the results presented in Fig. 4 suggest that AA has contributed to the change in summer circulation over Greenland.

- It would be interesting to see – also considering the analysis carried out in Figure 5 – in which month the sinuosity/GBI trends are significant and in which not. Indeed, it is shown that there is a global trend in JJA. However, the analysis presented later on seems to support that the direct influence of spring SCE on summer sinuosity is significant mostly in July (Figure 5bf). Consequently, the results suggest that the trend of JJA ascribable mainly to the trend in July. Is this confirmed by the single month trend analysis?

We have included the linear trends in the GBI and Sag for each summer month separately in the newly added Supplementary Fig. 1. The results show that July and August are the months with the strongest increase in both metrics.

Our analysis presents the decline in spring SCE as one contributing factor. That August has seen a comparable increase in both sinuosity and GBI clearly supports the idea that there are other factors at play. Previous work has also pointed to the reduction in sea ice as a possible contributor. We have edited the closing paragraph of our discussion to acknowledge this and included relevant references.

- I am not convinced by the explanation regarding the discrepancies between GBI and PH03. Indeed, blocking over Greenland is characterized by a very strong overturning, and such indices have been shown to capture very well the nature of the cyclonic wave breaking up there. My speculation is that the comparison you are doing is biased by the choice of using the original 1D index from Pelly and Hoskins (2003), rather than a 2D extensions as the one developed by Berrisford et al. (2007) and used by Woollings et al. (2008) or Masato et al. (2011). Indeed, using the single latitude based on EKE can be a very good metric, but it partially misses high latitude blocking as GB, since it passes too south of Greenland (see PH03 Figure 3). I encourage the authors to re-do the computation defining a 2D blocking index and then using a box-average over Greenland to define the presence of blocking on seasonal basis: I am quite convinced that this correlation will be largely increased. If they do not want to use theta-on-PV, a similar result should come out if you use a reversal index based on Z500 as the Scherrer et al (2004).

The reviewer makes an excellent point regarding the limitations of the original 1D blocking metric of Pelly and Hoskins (2003). We note that we did apply a modified Pelly and Hoskins index that considers seasonal variation in the EKE, which results in a more poleward JJA central blocking latitude that has been shown to better capture summer Greenland blocking (see Wachowicz et al., 2021). This supports the reviewer's reasoning that fully removing the latitude constraint by using a 2D extension would bring the two blocking metrics into closer agreement; however, there is also reason to believe that the GBI is better suited for this manuscript's examination of the mechanisms behind the shift in summer atmospheric circulation over Greenland—According to Woollings et al. (2018), even 2D variants of reversal-based metrics can fail to capture the recent positive trend in summer Greenland that is evident when using anomaly-based metrics such as the GBI.

Given the primary objective of this effort was to explore potential mechanisms behind the rise in Greenland blocking (which has largely been established using the GBI) and not to compare the performance of different blocking metrics, we have elected to follow the suggestion of Reviewer 1 and exclude the Pelly and Hoskins index results from the manuscript.

Minor Comments

L47: Please consider that NAO and Greenland blocking are simply two facets of the same dice, as shown by Woollings et al. (2008)

Thank you for highlighting this reference. We invoked this interpretation of the NAO in our discussion of the results in section 2.1. We feel this perspective of the NAO nicely ties together our results regarding the relationship between the NAO, GBI and Sinuosity.

L56: A consistent analysis of GB in both models and reanalysis can be found also in Davini and D'Andrea (2020), which showed that none of the CMIP5/CMIP6 models is able to replicate the trend.

Thank you for bringing this work to our attention. We have included it in the citations for the referenced sentence on line 56.

L120: is the standard deviation that increases or the average value of the timeseries that increase by 0.33 standard deviation? The standard deviation is computed over the entire period? Please clarify.

The average value of the time series increased by 0.33 standard deviation (0.28 SD after extending the analysis through 2022). We have made edits to clarify that the aggregate sinuosity is a standardized time series, such that the change with time is measured in units of standard deviation.

L121: which percentile you are discussing here? It is a bit unclear how do you measure your statistics here: at first, I assumed you have a single average value of Sag per season, which provide you a trend, but this does not seem to be the case. Do you mean that you compute percentiles on the each year daily Sag values and then assess their trend? If this is the case, please clarify the text since as it is written it is not obvious.

Thank you for pointing out the ambiguity here. To clarify, you are correct that trends that are reported first describe the seasonal mean Sag. We also calculated a range of percentile values for each season and year from the daily Sag values. This gives use, for example, the 90th percentile daily JJA Sag for each year. From this performed a linear regression for each percentile to see how the trends might vary across the distribution. We have made edits to clarify this in the second paragraph of section 2.1.

L126: please use a reference also for the GBI

Thank you for highlighting this omission. We have added references here for the GBI.

L134: are correlation in Figure 2b detrended? Please clarify that the trend has an impact on the correlations.

The correlation coefficients in Figure 2b describe the raw timeseries (i.e., not detrended). Thank you for highlighting the fact that this was not clear in our writing. We have made edits to section 2.1 and the figure caption to more precisely describe these results.

L160: Do you have any speculation while the fit tend to bend in favor of higher GBI for a lower sinuosity? Also, please keep in mind that NAO-GB relationship are not strictly linear, as shown by Woollings et al. (2010)

If we understand correctly, the reviewer referring to the fact that the best-fit line does not perfectly follow the one-to-one line in Fig 3. This could also be reflection of the situation where the lowest GBI values are accompanied by relatively higher sinuosity. Our interpretation is that this is because wavier atmospheric states do not always produce anomalous ridging over Greenland. While these results would suggest that wavy flows more frequently translate to anomalously anticyclonic conditions in the region, high sinuosity could also indicate the presence of a trough within the GBI domain.

Thank you again for highlighting this perspective on the NAO. We have incorporated it into our discussion at the end of this paragraph.

L170: I wonder how you can compute sinuosity at 80N: if you have a isolines which crosses the pole, how this is evaluated? Is this Sag still?

The results described here refer to sinuosity that has been calculated as a function of latitude—not Sag. The procedure for calculating sinuosity as a function of latitude is described briefly in section 4.1 and in more detail in Vavrus et al. (2017).

All sinuosity measures described in the manuscript are calculated for the regional domain shown in Fig. 1. An isoline that crosses the pole would exit the domain and the computed sinuosity would describe the only the portion of the isoline that is contained within the domain. Fig 1b provides a visual example for the 5500 hPa isoline, which is a more complex case of an isoline near the pole.

L194-196: this comment on the trend-induced correlation should go when discussing Figure 2.

We have edited our discussion of Fig. 2 in section 2.1 to ensure that the influence of the shared long-term trend is noted in both instances.

L224: I wonder what the correlation in Figure 5c,f means: a positive correlation between February SCE over North America and August GBI seems a bit counterintuitive. Since it is a quite surprising result, it would be nice if the authors can say something about it.

We agree with the reviewer that it is difficult to think of a mechanism that would link high North American SCE in February to above-normal Greenland blocking in August. This result seems even less plausible given the strong temporal autocorrelation that typifies seasonal snow cover progression and the fact that the correlation the following month (March) is opposite in sign. That the direct relationship between North American SCE area and August GBI is isolated to February makes us inclined to believe that this is an isolated case of correlation and not causation. We have incorporated this line of thinking into the discussion of the lagged correlations in section 2.2.

L238: Figure 6a, not 8a

This figure reference on line 238 has been corrected.

L239: “the vertical (colors) and horizontal (arrows) components of the associated WAF and its horizontal divergence (stippling)”

We have incorporated these clarifications.

L280: Why do the authors prefer to use snow cover duration instead of a simple snow cover fraction? Or are actually the same thing?

The NOAA SCE CDR does not provide information on the fraction of snow cover within each grid cell. Rather, it is a gridded Boolean, snow/no snow time series. Since spring snow cover extent closely reflects the timing of annual snow melt (i.e., early vs late snow melt), we used snow cover duration to construct a non-Boolean time series at each grid cell that could be compared against variability in the GBI. We have added this reasoning to section 2.3.

L281: are these correlations detrended?

Both measures were detrended prior to taking the correlation. This is now noted in section 2.3.

L289: it would be very informative to have also sensible heat flux here, since it is the driving mechanism that could sustain the wave activity flux shown in Figure 6. ERA5 data are available in this sense.

We have added companion panels illustrating the sensible heat flux response to Fig. 6.

L289: what is defined as “spring” SCE? Is this May North America SCE? Please clarify (also in Figure 8 caption).

We have edited the Fig. 8 caption and L289 to more precisely refer to May.

L316-326: Honestly, I am not getting the added value of Figure 9. I would suggest that the authors remove it.

This figure does provide additional information beyond what is presented in Fig. 6 by showing the westward tilt of the eddy geopotential height anomalies. We feel this is key to understanding the Rossby wave activity in July and have made edits to sections 2.2 and 2.3 to more effectively convey this in the manuscript.

The reviewer’s comment makes it clear that the material in Fig. 9 was likely not presented in the most effective manner. We have appended what was previously Fig. 9 as what is now Fig. 6c to more clearly show that it builds upon the WAF material.

L338: as discussed in the main comments, I think this claim is a bit too strong. As long as I see from Figure 5 AA is still increasing in recent years, while the GB trend is weakening.

To clarify, it appears the reviewer is referring to Fig. 4 here. After implementing the recommendations of Reviewer 1, Fig.4 shows that the trend in the meridional temperature gradient at ~60°N (Fig. 4c) that is accompanied by low p-values (Fig. 4d) levels off for 25-year regression windows centered around 2008 and later. This weakening of the meridional temperature gradient trend occurs in step with a weakening of the zonal wind and sinuosity trends at the same latitude. The correspondence between these variables is consistent with the mechanism linking AA to more persistent weather extremes presented in Francis and Vavrus (2012); however, it is not our intention to imply that AA is the only factor at play. This is the

point that we tried to make on L388. We acknowledge that this intention may not have been clear, and have softened our language accordingly.

L340-341: Actually this is not completely true: wave breaking can be cyclonic, and thus oriented over Greenland, or anticyclonic, and thus producing the most common European blocking. There are not so much different alternative, so that cyclonic wave breaking almost always occur over Greenland, as shown by multiple studies on atmospheric blocking (the already referenced Woollings et al. 2018 or Davini and D'Andrea 2020 are good examples). Please rephrase.

This was meant to be an a priori statement about the sinuosity index. In other words, before considering the influence of the local dynamics. A positive trend in sinuosity, on its own, does not inform on whether there is a tendency for more pronounced ridging over a given location. We have rephrased this sentence so that intention is clearer.

L366-369: I am not understanding why there is a mention to Omega block here. The geopotential height anomalies in Figure 5 show a reversal of the flow with negative anomalies over North Atlantic, which could remind even a Rex block. I would remove the reference to the Omega block here which seem to ma bit forced.

The reviewer is correct that this is not an essential point. We have removed this paragraph.

L370: "retreating North American SCE".

This correction has been made.

Figure 2b: rather than using the colors for highlighting the magnitude of the correlation, it would be more useful to use it to mark significant/non-significant correlations.

After removing the PH03 results, all correlations in the correlation matrix are significant. We have noted this in the figure caption.

Figure 5. Please use lettering for panels and not row/columns.

We have made this change to Fig. 5

REVIEWERS' COMMENTS

Reviewer #1 (Remarks to the Author):

The authors have done much to make their statistical analyses more clear. In particular, changing from an expanding window regression to a rolling window regression to produce figure 4 is an improvement.

Reviewer #2 (Remarks to the Author):

I appreciate the efforts made by the author to rerun CESM models and properly address the raised questions. I am satisfied with the revisions but still have several suggestions for the authors.

1. As the authors noted, the North Atlantic circulation responses is one-month earlier than that in the observation when the model was forced by May SCE anomalies. A more appropriate experimental design would be a persistent SCE forcing prescribed in the model from April to July to drive the delayed summertime circulation anomalies. The question would be the timing of delayed response to preceding surface condition forcings. This is only a comment, not an obligation to repeat the model experiments.

2. Several important literatures are ignored by the authors when review the progress of snow cover and soil moisture effects. For instance:

Lines 91-94 and 326-328: Spring SCE influences the summertime atmospheric circulation through the "memory effect" of soil moisture:

Shinoda, M., 2001: Climate memory of snow mass as soil moisture over central Eurasia. *J. Geophys. Res.*, 106, 33 393–33 403, doi:10.1029/2001JD000525.

Zhang, R.N., R.H. Zhang, and Z. Zuo, 2017: Impact of Eurasian spring snow decrement on East Asian summer precipitation. *Journal of Climate*, 30, 3421–3437, doi: 10.1175/JCLI-D-16-0214.1.

Lines 94-95: which can then influence the generation of Rossby waves:

Petoukhov, V., Rahmstorf, S., Petri, S. & Schellnhuber, H. J. Quasiresonant amplification of planetary waves and recent Northern Hemisphere weather extremes. *Proc. Natl Acad. Sci.* 110, 5336–5341 (2013).

Zhang, R.N., C. Sun, J. Zhu, R.H. Zhang, and W. Li, 2020: Increased European heat waves in recent decades in response to shrinking Arctic sea ice and Eurasian snow cover. *npj Climate and Atmospheric Science* 3, 7. <https://doi.org/10.1038/s41612-020-0110-8>.

Cohen, J., and D.Rind, 1991: The effect of snow cover on the climate. *J. Climate*, 4, 689–706, doi:10.1175/15200442(1991)004, 0689

REVIEWERS' COMMENTS

Reviewer #1 (Remarks to the Author):

The authors have done much to make their statistical analyses more clear. In particular, changing from an expanding window regression to a rolling window regression to produce figure 4 is an improvement.

Thank you for helping us to clarify and strengthen our statistical analyses.

Reviewer #2 (Remarks to the Author):

I appreciate the efforts made by the author to rerun CESM models and properly address the raised questions. I am satisfied with the revisions but still have several suggestions for the authors.

1. As the authors noted, the North Atlantic circulation responses is one-month earlier than that in the observation when the model was forced by May SCE anomalies. A more proper experimental design would be a persistent SCE forcing prescribed in the model from April to July to drive the delayed summertime circulation anomalies. The question would be the timing of delayed response to preceding surface condition forcings. This is only a comment, not an obligation to repeat the model experiments.

We thank the reviewer for this feedback. We look forward to performing a more extensive analysis of GCM representation of this snow-atmosphere coupling as part of future work.

2. Several important literatures are ignored by the authors when review the progress of snow cover and soil moisture effects. For instance:

Lines 91-94 and 326-328: Spring SCE influences the summertime atmospheric circulation through the "memory effect" of soil moisture:

Shinoda, M., 2001: Climate memory of snow mass as soil moisture over central Eurasia. *J. Geophys. Res.*, 106, 33 393–33 403, doi:10.1029/2001JD000525.

Zhang, R.N., R.H. Zhang, and Z. Zuo, 2017: Impact of Eurasian spring snow decrement on East Asian summer precipitation. *Journal of Climate*, 30, 3421–3437, doi: 10.1175/JCLI-D-16-0214.1.

Lines 94-95: which can then influence the generation of Rossby waves:

Petoukhov, V., Rahmstorf, S., Petri, S. & Schellnhuber, H. J. Quasiresonant amplification of planetary waves and recent Northern Hemisphere weather extremes. *Proc. Natl Acad. Sci.* 110, 5336–5341 (2013).

Zhang, R.N., C. Sun, J. Zhu, R.H. Zhang, and W. Li, 2020: Increased European heat waves in recent decades in response to shrinking Arctic sea ice and Eurasian snow cover. *npj Climate and Atmospheric Science* 3, 7. <https://doi.org/10.1038/s41612-020-0110-8>.

Cohen, J., and D.Rind, 1991: The effect of snow cover on the climate. *J. Climate*, 4, 689–706, doi:10.1175/15200442(1991)004, 0689

Thank you for highlighting these references. We have added them to the manuscript.